# Ribosome profiling reveals a functional role for autophagy in mRNA translational control

Juliet Goldsmith [1], Timothy Marsh [1], Saurabh Asthana[2,3], Andrew M. Leidal[1], Deepthisri Suresh[1], Adam Olshen[2,3] & Jayanta Debnath [1,3✉]

Autophagy promotes protein degradation, and therefore has been proposed to maintain amino acid pools to sustain protein synthesis during metabolic stress. To date, how autophagy influences the protein synthesis landscape in mammalian cells remains unclear. Here, we utilize ribosome profiling to delineate the effects of genetic ablation of the autophagy regulator, ATG12, on translational control. In mammalian cells, genetic loss of autophagy does not impact global rates of cap dependent translation, even under starvation conditions. Instead, autophagy supports the translation of a subset of mRNAs enriched for cell cycle control and DNA damage repair. In particular, we demonstrate that autophagy enables the translation of the DNA damage repair protein BRCA2, which is functionally required to attenuate DNA damage and promote cell survival in response to PARP inhibition. Overall, our findings illuminate that autophagy impacts protein translation and shapes the protein landscape.

[1] Department of Pathology, University of California San Francisco, San Francisco, CA 94143, USA. [2] Department of Epidemiology and Biostatistics, University of California San Francisco, San Francisco, CA 94158, USA. [3] Helen Diller Comprehensive Cancer Center, University of California San Francisco, San Francisco, CA 94158, USA. ✉email: jayanta.debnath@ucsf.edu

 **1**

Autophagy is a cellular recycling system that degrades proteins and organelles by delivery to the lysosome and promotes cell survival in response to metabolic and oxidative stress[1]. As such, autophagy is rapidly upregulated, both post-translationally and transcriptionally, during nutrient starvation[2]. At the same time, mRNA translation is tightly regulated by the metabolic state of the cell. The Rag complex senses lysosomal amino acid levels and signals through mTORC1 to regulate cap-dependent mRNA translation[3,4]. Upon amino acid starvation, reduced mTOR signaling attenuates global cap-dependent mRNA translation, while concurrently inducing autophagy[5,6]. Accordingly, in starving cells and tissues, autophagy-mediated recycling of amino acids is proposed to sustain residual translation of proteins, in particular those necessary for survival and metabolic adaptation during starvation or stress. In support, studies in *Saccharomyces cerevisiae* demonstrate that autophagy is crucial to maintain protein synthesis during nitrogen starvation[7]. However, in mammalian cells, it remains unclear whether autophagy similarly impacts protein synthesis, either in nutrient replete or starvation conditions.

Here, we utilize ribosome profiling to dissect how the autophagy pathway impacts the mRNA translation landscape, both at baseline and in response to starvation. We uncover indirect roles for autophagy in regulating the translation of specific mRNAs, distinct from tuning protein synthesis rates in mammalian cells. In contrast to previous results from *Saccharomyces cerevisiae*, genetically abolishing autophagy in mammalian cells does not dramatically impair amino acid recycling, nor does it globally suppress protein synthesis rates, or cap-dependent or internal ribosome-entry site (IRES)-dependent mRNA translation during nutrient stress. Instead, autophagy inhibition leads to specific changes in the translation of a group of mRNAs involved in DNA repair, centrosome clustering and cell-cycle control. To further understand the mechanisms controlling autophagy-regulated mRNA translation, we specifically focus on how autophagy enables translation of the DNA damage repair gene *Brca2*. Loss of autophagy results in diminished levels of BRCA2 and increased DNA damage, which can be partly rescued upon ectopically enforcing BRCA2 expression. Furthermore, the 5′UTR sequence and structure of Brca2 is an important determinant of the sensitivity to translational control in response to autophagy inhibition. We propose that autophagy-dependent regulation contributes to the efficient translation of proteins necessary for DNA damage repair and cell-cycle fidelity.

## Results

**Autophagy loss does not affect translation rates**. ATG12 is an autophagy regulator required for the elongation of the double-membrane structure during autophagosome formation[8,9]. To limit the effects of long-term adaptation due to the lack of autophagy in mammalian cells, we created a cell culture model for rapid and efficient *Atg12* deletion. SV40 large T antigen immortalized mouse embryonic fibroblasts (MEF) homozygous for *Atg12* floxed alleles[10], and heterozygous for the Cre[ER] allele driven from the ubiquitous Cag promoter (*Atg12*[f/f]; *Cag-Cre*[ER]), were treated with 4-hydroxytamoxifen (4OHT), resulting in *Atg12* ablation and robust autophagy inhibition. Within 2d, the null allele was detectable by PCR (Supplementary Fig. 1a), and after 5d, no detectable Atg12 protein was found by immuno-blotting. Lipidation and lysosomal turnover of LC3 (LC3-II) was profoundly attenuated in *Atg12*[KO] cells, resulting in the accumulation of the autophagy cargo receptor, p62/SQSTM1 (Supplementary Fig. 1b). For subsequent studies, we analyzed *Atg12*[KO] cells at 5d following 4OHT treatment.

First, we assessed the effect of autophagy loss on overall global protein synthesis rates, using a [35]S methionine incorporation assay and a puromycin incorporation assay. Cells were starved for 2 h in Hanks buffered saline solution (HBSS), a serum free, amino acid-free saline solution containing glucose. This brief starvation period induces autophagy but precedes major transcriptional changes associated with starvation[11] (Supplementary Fig. 1c). Interestingly, we consistently observed robustly increased [35]S methionine incorporation during HBSS starvation, in contrast to the removal of single nutrients, such as glucose or glutamine (Fig. 1a, b). However, upon alternatively assaying protein synthesis rates using puromycin incorporation, we observed a trending decrease in HBSS-starved cells (Fig. 1c, d). Most importantly, upon evaluating the effects of genetic autophagy inhibition on the rates of de novo protein synthesis, we found no significant differences in either [35]S methionine incorporation or puromycin incorporation between *Atg12*[KO] and control (*Atg12*[f/f]) cells, in either fed or starved conditions (Fig. 1b, d). Similar results were observed in a broader array of immortalized MEFs lacking various autophagy regulators, including ATG12, ATG5, ATG7, or ATG3 (Supplementary Fig. 1d) as well as in primary MEFs lacking ATG12 (Supplementary Fig. 1e). Hence, the genetic loss of autophagy does not acutely impact the rate of de novo protein synthesis under either fed or starved conditions in mammalian cells, in contrast to previous results in *Saccharomyces cerevisiae*[7].

The availability of translation initiation factors or variant isoforms can regulate the rate of translation and impact which mRNAs are translated[12–14]. Although phosphorylated initiation factor 2-alpha (p-eIF2α), which represses cap-dependent global translation[15], was slightly increased, these changes were not statistically significant (Fig. 1e, f). There was no significant difference in the ratio of IRES-dependent to cap-dependent translation between *Atg12*[f/f] and *Atg12*[KO] cells using well-characterized viral IRES motifs from cricket paralysis virus (CrPV) (Fig. 1g) and Hepatitis C virus (HCV) (Supplementary Fig. 1f). In addition, using a m[7]GTP cap-pulldown assay, we observed no significant differences in the cap-binding abilities of key initiation factors eIF4E, eIF4G1, or their variants eIF4E2 or eIF4G2 between *Atg12*[f/f] and *Atg12*[KO] cells (Fig. 1h), nor any differences in the binding of the inhibitory factor 4E binding protein (4EBP1) to the m[7]GTP cap (Fig. 1i).

**Atg12 loss minimally impacts mTORC1 activity**. In parallel, we measured whether *Atg12* deletion impacts intracellular free amino acid levels, and found minimal differences between *Atg12*[f/f] and *Atg12*[KO] MEFs in both nutrient-rich conditions and following HBSS starvation for up to 2 h (Fig. 2a, Supplementary Fig. 2a). While glutamine may have lower levels in *Atg12*[KO] cells (Supplementary Fig. 2b), only glycine was significantly decreased in *Atg12*[KO] cells compared with controls grown in nutrient-rich full media conditions (Supplementary Fig. 2c), and upon starvation, the only amino acid lost more rapidly in *Atg12*[KO] cells was glutamic acid (Supplementary Fig. 2d). Notably, *Atg12*[KO] cells exhibited increased levels of oxoproline during starvation (Supplementary Fig. 2e), suggesting low glutamine levels may be due to reduced extracellular glutamine uptake through the gamma-glutamyl cycle. Notably, essential amino acids, including the branched chain amino acids (leucine, isoleucine, and valine), serine and threonine all exhibited higher measured levels in *Atg12*[KO] cells compared with controls at baseline (Supplementary Fig. 2f–j). Arginine is not discernible by this technique, however hydroxylamine levels were higher in the *Atg12*[KO] starved cells, suggesting arginine metabolism in the autophagy-deleted cells may be enhanced (Supplementary Fig. 2k). Although autophagy is proposed to sustain de novo protein synthesis by recycling

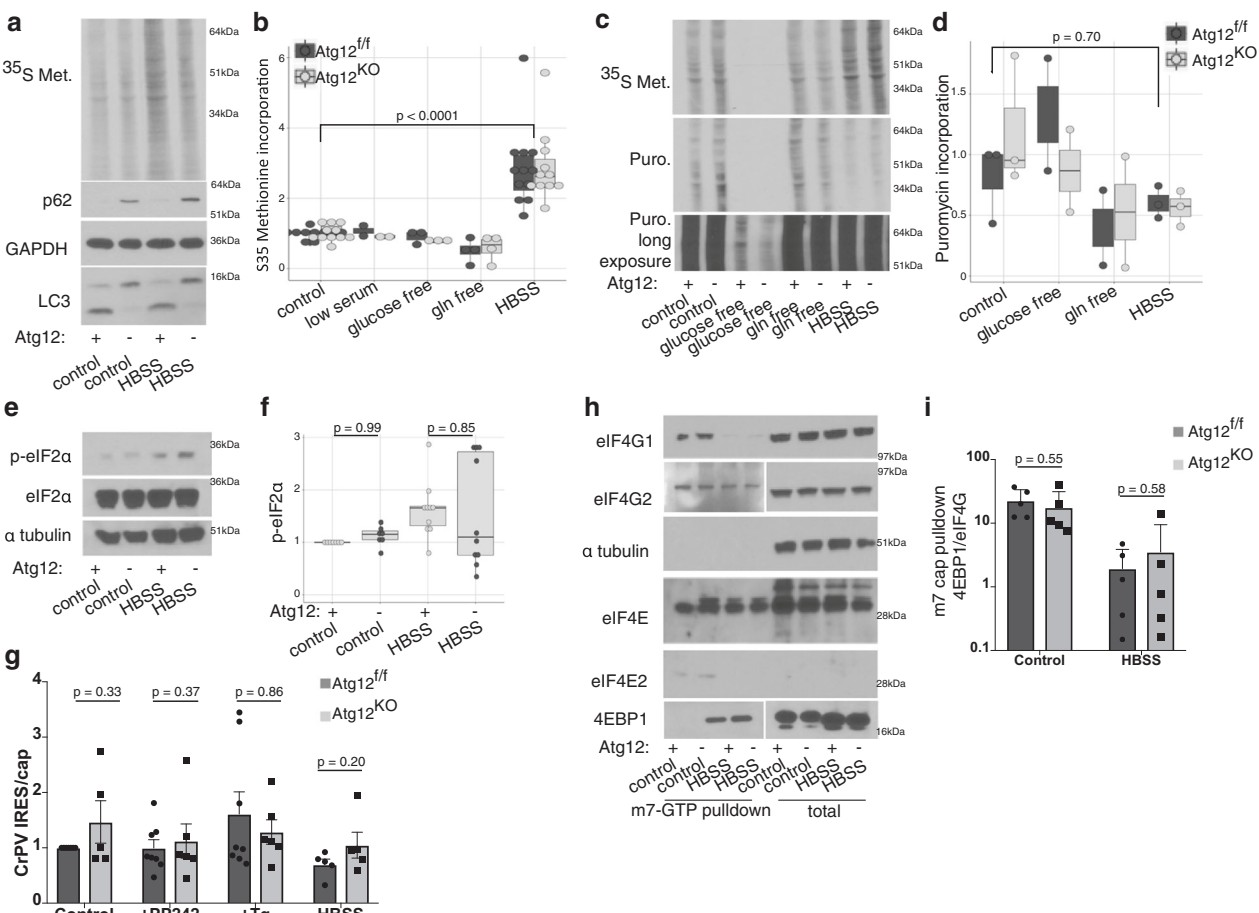

**Fig. 1 Minimal effects of ATG12 genetic deletion on global translation. a** Representative $^{35}$S methionine incorporation autoradiogram from *Atg12*$^{f/f}$ and *Atg12*$^{KO}$ MEFs in control media or following 2 h HBSS starvation. p62/SQSTM1 and LC3 immunoblotting on the same blot is shown below. **b** *Atg12*$^{f/f}$ or *Atg12*$^{KO}$ MEFs were switched to either control, low serum, glucose free, or glutamine-free media, or HBSS for 2 h. Cells were labeled with $^{35}$S methionine for 30 min prior to lysis. Relative $^{35}$S methionine incorporation rate is quantified, shown as a boxplot with dotplot overlay for each biological replicate, normalized to loading control. Statistical significance was assessed by ANOVA with Tukey's HSD post hoc test, $p = 1.0$ for *Atg12*$^{f/f}$ cells compared with *Atg12*$^{KO}$ cells, in both control and HBSS. **c**, **d** *Atg12*$^{f/f}$ and *Atg12*$^{KO}$ MEFs were cultured in control, glucose free, or glutamine-free media or HBSS for 2 h. Addition of $^{35}$S methionine and puromycin was included for 30 min prior to lysis. **c** Representative autoradiogram and immunoblot and (**d**) relative puromycin incorporation rate is quantified, displayed as a boxplot including each biological replicate, normalized to loading control; $p > 0.95$ for *Atg12*$^{f/f}$ cells compared with *Atg12*$^{KO}$ cells, in both control media and HBSS. **e**, **f** *Atg12*$^{f/f}$ and *Atg12*$^{KO}$ MEFs in control media or following 2 h HBSS starvation were lysed and immunoblotted for markers of cap-dependent protein translation inhibition (p-eIF2α). **e** Representative immunoblots are shown. **f** Relative protein levels of p-eIF2α were quantified, normalized to loading control, and shown as a boxplot including each biological replicate. Statistical analysis by ANOVA with Tukey's HSD post hoc test. **g**, **h** Protein lysate from *Atg12*$^{f/f}$ and *Atg12*$^{KO}$ MEFs in control media or following 2 h HBSS starvation was subject to pulldown with γ-amino-phenyl-m7-GTP cap analog conjugated to agarose beads. **g** A representative immunoblot is shown and (**h**) 4EBP1 relative to eIF4G1 levels are quantified (mean + SD, $n = 3$ independent biological replicates), $p$ value by $t$ test. **i** Quantification (mean + SEM, $n = 5$ independent biological replicates, $p$ value by $t$ test) of Cricket paralysis virus IRES translation, normalized to cap translation rates. Cells were treated with PP242 (2 μM for 1 h) to inhibit mTORC1, and Thapsigargin (Tg, 1 μM for 1 h) to induce IRES-mediated translation (additional data in Supplementary Fig. 1f).

amino acids, these results indicate that intracellular levels of most amino acids remain intact in autophagy-deficient cells following short-term nutrient starvation.

Amino acid levels affect the activity of mTORC1, which regulates the translation of mRNAs containing terminal oligo-pyrimide (TOP) motifs including translational machinery[16], and is considered a master regulator of cell growth and protein synthesis[17]. Therefore, we investigated the effects of autophagy inhibition on mTORC1. Downstream markers of mTORC1 activation, phosphorylation of 4EBP1 at Ser65 and ribosomal protein S6 at Ser240 and 244, demonstrated that mTORC1 activity was robustly inhibited following HBSS starvation in both autophagy competent and deficient cells (Fig. 2b–e)[18]. No significant differences in key markers of mTORC1 activity were

observed between *Atg12*$^{f/f}$ and *Atg12*$^{KO}$ cells, either in full media or upon HBSS starvation (Fig. 2c, d).

Overall, these results indicate that the genetic loss of autophagy in both nutrient replete and short-term starvation conditions does not impact mTORC1 signaling or global protein synthesis.

**Autophagy modulates the translation of specific mRNAs.** To more thoroughly understand the impact of autophagy on mRNA translation, we employed ribosome profiling (RP), a sensitive and unbiased technique to identify the changes in the rate of translation of all expressed mRNAs in the context of autophagy deficiency[19]. Briefly, translating ribosomes are fixed onto mRNAs by treatment with cycloheximide so that ribosome protected

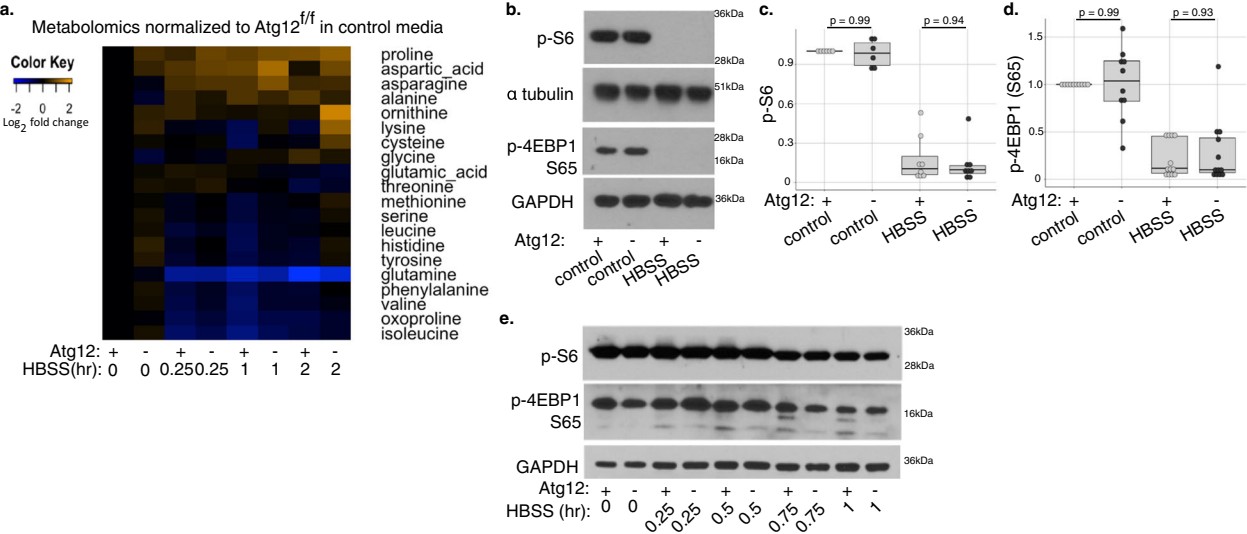

**Fig. 2 Amino acid levels and mTORC1 activation not impaired by loss of ATG12. a** Changes in intracellular metabolite levels detected by GC-TOF with MTBSTFA ($n = 4$ biologically independent replicates), in *Atg12*[f/f] and *Atg12*[KO] MEFs in control media or following HBSS starvation for the indicated times. Fold change is relative to *Atg12*[f/f] in control media. **b–d** *Atg12*[f/f] and *Atg12*[KO] MEFs in control media or following 2 h HBSS starvation were lysed and immunoblotted for markers of mTORC1 signaling (p-S6, p-4EBP1). **b** Representative immunoblot is shown. Relative protein levels of (**c**) p-S6 and (**d**) p-4EBP1 were quantified, normalized to loading control, and shown as a boxplot with dotplot overlay for each biological replicate. Statistical analysis by ANOVA with Tukey's HSD post hoc test was performed and *p* values for *Atg12*[f/f] samples compared with *Atg12*[KO] samples in control media or HBSS are shown. **e** Representative immunoblot for markers of mTORC1 signaling (p-S6, p-4EBP1) in *Atg12*[f/f] and *Atg12*[KO] MEF protein lysate following a timecourse of HBSS starvation.

footprints (RPFs) can be isolated, amplified, deep-sequenced, mapped to the transcriptome and normalized to total mRNA levels measured by parallel RNA sequencing. Because the ribosome counts are normalized to mRNA levels, confounding gene expression changes are factored into the statistical analysis, hence allowing evaluation of ribosome occupancy, and by inference, translation rates at a particular snapshot in time.

*Atg12*[f/f] and *Atg12*[KO] cells were compared to each other in full media conditions and after 2 h HBSS starvation. Verifying experimental quality and reproducibility between replicates, we found generally equal and low levels of contaminating rRNA reads, RPF versus mRNA count plots were similar, and found a statistically significant correlation (Pearson's) of both the raw values of RNA and RPF counts and the calculated *p* values (Supplementary Fig. 3a, b, Supplementary Table 1). Minimal changes in the numbers of RPF counts per mRNA were found between *Atg12*[f/f] and *Atg12*[KO] cells, while RPF counts in starved versus fed *Atg12*[f/f] cells decreased over all biological replicates (Fig. 3a–c). Analysis of the fold change of RPF counts vs. fold change of mRNA counts per gene revealed general changes in the transcriptional and translational landscape (Fig. 3d–f, Supplementary Fig. 3c). Statistical significance of ribosome occupancy changes normalized to mRNA changes was assessed at the gene level using Babel[20,21]; Supplementary Data 2 lists the 30 most significant genes from each comparison. ATG12 affected ribosome occupancy on a small subset of mRNAs, both positively and negatively.

We grouped genes with significant ribosome occupancy changes between *Atg12*[f/f] and *Atg12*[KO] cells into two cohorts: those exhibiting increased ribosome occupancy in *Atg12*[KO] compared with *Atg12*[f/f] (Fig. 3g, $p < 0.01$), and those exhibiting reduced ribosome occupancy in *Atg12*[KO] cell compared with *Atg12*[f/f] (Fig. 3h, $p < 0.005$). Among the cohort of mRNAs exhibiting reduced ribosome occupancy in *Atg12*[KO] cells, gene ontology (GO) analysis revealed an observed enrichment of genes involved in cell-cycle control and chromosome organization

(Supplementary Fig. 3d). In contrast, no significant enrichment in biological processes were evident in the cohort displaying increased ribosome occupancy in *Atg12*[KO] cells. Notably, this GO enrichment correlated with slowed cell-cycle progression in autophagy-deficient cells. *Atg12*[KO] cells exhibited slower growth rates compared with the *Atg12*[f/f] cells in full media (Supplementary Fig. 3e) and higher percentage of phospho-Histone H3 (pH3) positive cells (Supplementary Fig. 3f), indicating slower progression through mitosis. We observed a significantly decreased percentage of cells in G1 in an unsynchronized population (Supplementary Fig. 3g). We propose that the translation of cell-cycle control mRNAs may represent an important consequence of enhanced autophagic flux observed during early mitosis and S phase[22].

**Autophagy enables the translation of Brca2**. Several genes with significantly lower ribosome occupancy in *Atg12*[KO] cells regulate the DNA damage repair response, a process previously linked to autophagy[23]. We focused on the function of autophagy to enable the translation of Brca2. BRCA2, which is commonly deleted in heritable breast cancer, functions in DNA double strand break repair and centrosome clustering. We labeled newly synthesized protein in cells with azidohomoalanine (AHA), a methionine analog that can be conjugated to biotin, and pulled down BRCA2 to monitor the rate of label incorporation. We observed impaired label incorporation in the *Atg12*[KO] cells compared with control cells (Fig. 4a, b), demonstrating a reduced rate of BRCA2 synthesis in *Atg12*[KO] cells and consistent with reduced ribosome occupancy on Brca2 mRNA.

The decrease in the rate of BRCA2 production in *Atg12*[KO] cells correlated with reduced BRCA2 protein levels compared with controls in fed and starved conditions (Fig. 4c, d). In addition, CRISPR engineered HEK293T cells lacking ATG12 exhibited lower steady state BRCA2 protein levels (Fig. 4e, f), indicating that autophagy-dependent control of BRCA2 protein levels is not

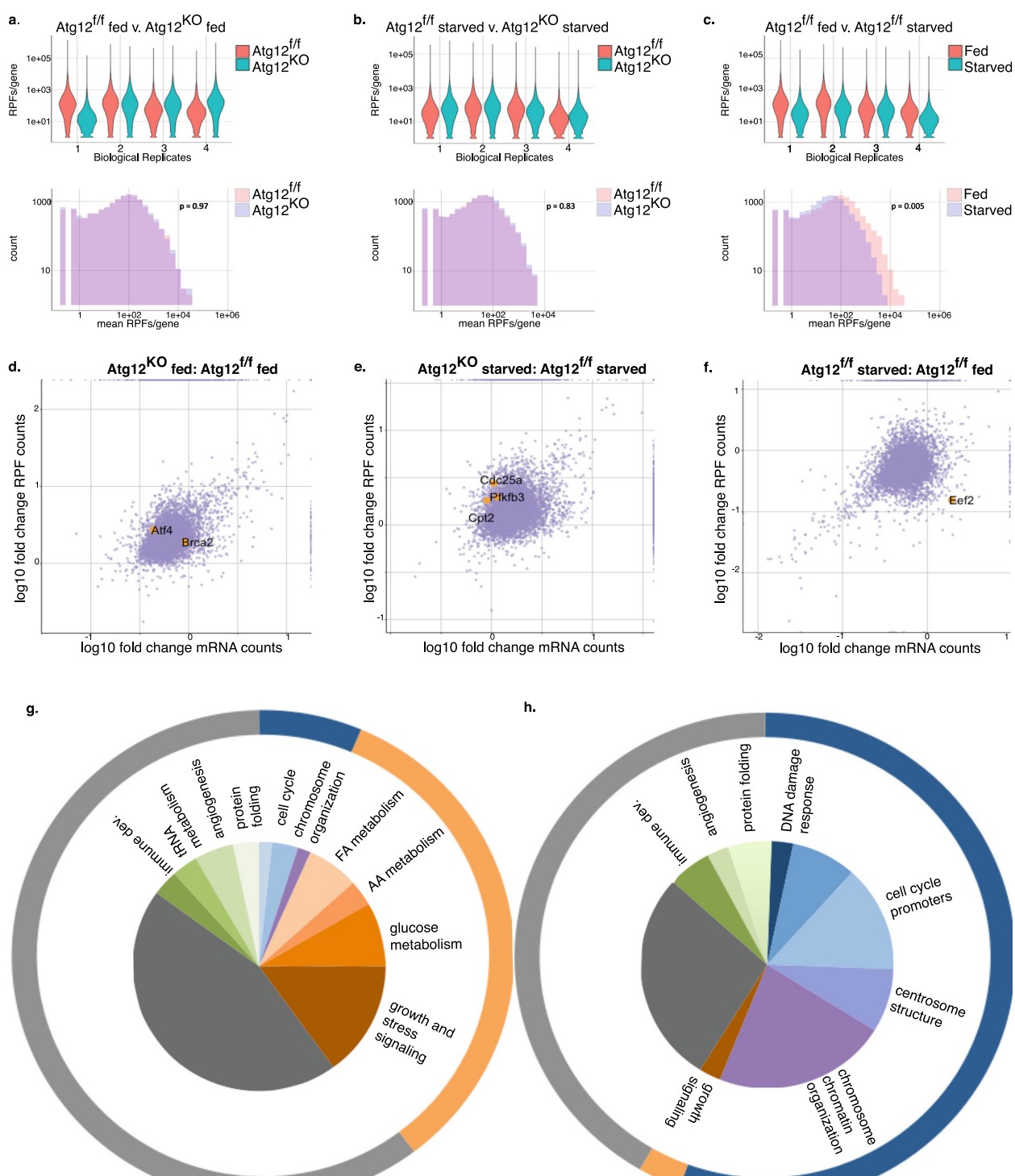

**Fig. 3 Ribosome profiling reveals that autophagy supports the translation of proteins involved in DNA damage response and cell-cycle control.**
**a–c** Violin plots of number of read counts of ribosome protected footprints (RPFs) per gene per biological replicate (above) and histogram of the mean of the number of read counts of ribosome protected footprints per gene (below) in (**a**) *Atg12*[f/f] and *Atg12*[KO] MEFs in control media (**b**) *Atg12*[f/f] and *Atg12*[KO] MEFs following 2 h HBSS starvation or (**c**) *Atg12*[f/f] MEFs in control media or following 2 h HBSS starvation, $p = 0.005$ by $t$ test. **d–f** Fold change of RPF counts versus fold change of mRNA counts. Labeled points in orange are mRNAs whose change in ribosome occupancy was significant, and protein level changes confirmed by immunoblotting (see Supplementary Fig. 3c). **g, h** Molecular functions of mRNAs whose ribosome occupancy is **g** increased ($p < 0.01$, $n = 36$ significant genes) or **h** decreased ($p < 0.005$, $n = 60$ significant genes) in *Atg12*[KO] cells versus *Atg12*[f/f] cells in either fed or starved conditions.

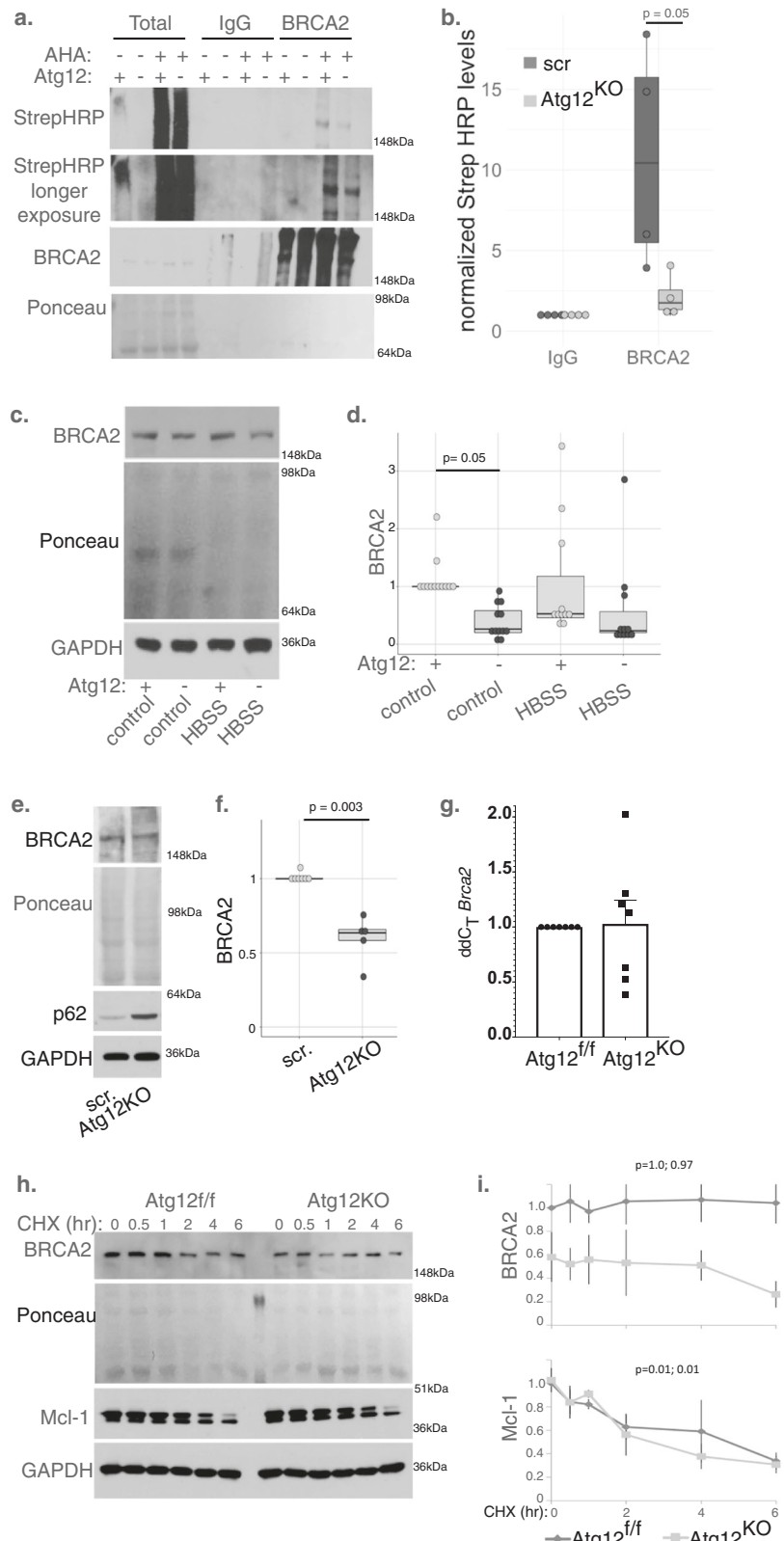

limited to fibroblasts. To assess whether reduced levels of BRCA2 arose from defective autophagy, we probed for BRCA2 levels in *Atg5*-deleted and Bafilomycin-treated MEFs and CRISPR engineered HEK293T cells lacking ATG7 or ATG14 (Supplementary Fig. 4a–f). Steady state BRCA2 levels were decreased autophagy-inhibited MEFs. ATG7- and ATG14-knockout HEK293T cells had variable expression of BRCA2; we hypothesize that a

compensatory upregulation of BRCA2 occurred to promote cellular health during the single-cell clone selection, or that due to variability, the statistical analysis was underpowered.

To confirm that decreased BRCA2 steady state protein levels were due to reduced translation, we assessed Brca2 transcription and BRCA2 stability and turnover in *Atg12*KO cells. We found no significant difference between *Atg12*f/f and *Atg12*KO MEFs in

**Fig. 4 Reduced BRCA2 protein translation in autophagy-deficient cells. a, b** Newly synthesized BRCA2 levels were assessed following 8 h AHA incorporation and BRCA2 immunoprecipitation from control or $Atg12^{KO}$ HEK293T cells. **a** Representative immunoblot is shown. **b** Quantification (boxplot with dotplot overlay for each independent biological replicate) of newly synthesized BRCA2 (StrepHRP levels), $p = 0.05$ by $t$ test. **c, d** Protein lysate was collected from $Atg12^{f/f}$ and $Atg12^{KO}$ MEFs in control media or following 2-h HBSS starvation. BRCA2 levels were measured by immunoblotting: **c** representative immunoblot is shown; **d** relative BRCA2 protein levels were normalized to loading control, and quantified shown as a boxplot with dotplot overlay for each biological replicate. Statistical analysis was performed by ANOVA with Tukey's HSD post hoc test ($p = 0.05$ for control conditions). Comparing $Atg12^{f/f}$ to $Atg12^{KO}$ in HBSS, one outlier (Dixon test, $p = 0.001$) was excluded from statistical analysis, $p = 0.05$ by $t$ test. **e, f** Protein lysate was collected from HEK293T cells with CRISPR deleted ATG12 and (**e**) immunoblotted for the indicated proteins; **f** relative BRCA2 protein levels normalized to loading control was quantified and shown as boxplot with dotplot overlay for each independent replicate, $p = 0.003$ by $t$ test. **g** $Brca2$ levels (mean ± SD, $n = 7$ independent biological replicates) in $Atg12^{f/f}$ and $Atg12^{KO}$ MEFs were measured by qPCR, with $Gapdh$ as the endogenous control. **h, i** $Atg12^{f/f}$ and $Atg12^{KO}$ MEFs following cycloheximide (100 μg/ml) treatment for time indicated to inhibit protein translation. **h** Representative immunoblots for BRCA2 and Mcl-1. **i** Quantification (mean ± SD, $n = 3$ independent biological replicates) of relative BRCA2 and Mcl-1 levels from immunoblotting, normalized to loading control. $p$ values for cyclohexamide treatment between $t = 0$ and 6 h were calculated using ANOVA with Tukey's post hoc test and are shown for $Atg12^{f/f}$ and $Atg12^{KO}$, respectively.

either Brca2 mRNA levels (Fig. 4g) or in BRCA2 protein stability or turnover following cycloheximide treatment (Fig. 4h, i), which did not impact autophagic flux (Supplementary Fig. 4g), indicating that the lower levels of BRCA2 were not due to changes in Brca2 expression levels, nor enhanced degradation of BRCA2. Overall, these results demonstrate efficient Brca2 translation and maintenance of normal BRCA2 protein levels requires an intact autophagy pathway.

**The 5′UTR of Brca2 directs autophagy-enabled translation.** The untranslated regions (UTRs) of mRNAs function as important regulators of translational efficiency; 5′UTRs can contain upstream open reading frames and motifs to interact with various RNA-binding proteins (RBPs), while the 3′UTR harbors RBP binding and miRNA motifs[24]. We interrogated whether the Brca2 5′ or 3′ UTR mediated translational control downstream of autophagy. A green fluorescent protein (GFP) reporter was transiently overexpressed in $Atg12^{f/f}$ or $Atg12^{KO}$ MEFs alone or flanked by the Brca2 5′UTR, 3′UTR, or both UTRs. GFP protein levels were decreased in the presence of the 5′UTR of Brca2, but not the 3′UTR, in $Atg12^{KO}$ compared with $Atg12^{f/f}$ cells, despite equivalent Gfp expression (Fig. 5a, Supplementary Fig. 5a). Moreover, utilizing nano-luciferase reporters to quantitatively measure the effects of Brca2 UTRs, we observed significantly decreased luciferase activity in the $Atg12^{KO}$ MEFs compared with $Atg12^{f/f}$ when the 5′UTR of $Brca2$ preceded luciferase (Fig. 5b). Therefore, the 5′UTR of Brca2 contains the region that mediates autophagy-dependent translation of this mRNA.

Remarkably, the 5′UTRs of the cohort of mRNAs exhibiting lower RP occupancy in $Atg12^{KO}$ cells had significantly lower folding energies compared with the 5′UTRs of a random sampling of mouse genes (Supplementary Fig. 5b). To adjust for length of the UTRs, the minimum free energy (MFE) within the 5′UTRs was predicted by RNALfold[25]; significant differences were detected between the two groups (Fig. 5c, Supplementary Fig. 5c, d). These results indicate that mRNAs whose translation efficiency is enhanced in autophagy competent cells possess above average 5′UTR secondary structure complexity. Indeed, Irf7, another hit from our ribosome profiling screen is notable for a complex 5′UTR secondary structure[26,27] (Supplementary Fig. 5e).

The secondary structure of the 5′UTR can slow, or prevent, translation via diverse molecular mechanisms[28]. Since mRNAs with complex secondary structures rely upon RNA helicases to facilitate the loading and reading of ribosomes[29,30], we investigated whether the helicase eIF4A1, part of the eIF4F initiation complex that recruits ribosomes to mRNA, exhibited impaired binding to Brca2 in $Atg12^{KO}$ cells. RNA immunoprecipitation (RIP) confirmed decreased interaction between endogenous eIF4A1 and Brca2 mRNA in $Atg12^{KO}$ versus $Atg12^{f/f}$ cells

(Fig. 5d, Supplementary Fig. 5f). Notably, RIP also revealed similarly trending decreases in interactions between eIF4A1 and mRNAs for Irf7 and Trp53, two other ribosome profiling hits with complex secondary structure in their 5′UTRs[26,27,31] (Supplementary Fig. 5g, h). Although the total protein levels of eIF4A1 were unchanged in $Atg12^{KO}$ cells (Supplementary Fig. 5i), cap-pulldown assays demonstrated reduced interaction between the endogenous eIF4A1 and the exogenous sepharose-coupled $m^7GTP$ cap in $Atg12^{KO}$ cells (Fig. 5e, f), suggesting the sequestration of eIF4A1 away from mRNAs in autophagy-deficient cells.

**Accumulation of p62 sequesters eIF4A1 from Brca2.** To further understand eIF4A1 sequestration in $Atg12^{KO}$ cells, we tested the interaction between eIF4A1 and known autophagy cargo receptors (ACRs), mediators of selective autophagy that accumulate upon autophagy inhibition. We observed increased co-location of eIF4A1 within puncta of the ACR p62/SQSTM1 in autophagy-deficient cells (Fig. 6a, b) and immunoprecipitation studies indicated that endogenous eIF4A1 interacts with endogenous p62/SQSTM1 in $Atg12^{KO}$ but not $Atg12^{f/f}$ cells (Fig. 6c, Supplementary Figs. 6a, 5f). In contrast, autophagy deficiency did not significantly enhance the interaction of eIF4A1 with NBR1, a similar ACR (Fig. 6b, c). p62/SQSTM1 depletion rescued the ability of eIF4A1 to interact with the cap, and overexpression of a mutant p62/SQSTM1 that cannot be degraded by autophagy (p62ΔLIR) reduced eIF4A1 binding to the cap (Fig. 6d, e). Furthermore, p62/SQSTM1 depletion was able to enhance eIF4A1 binding to Hnrnpc, whose translation is dependent on eIF4A1[32] (Supplementary Fig. 6b). However, p62/SQSTM1 knockdown was not sufficient to restore BRCA2 levels in $Atg12^{KO}$ cells (Fig. 6f–h). In addition, although eIF4A1 interaction with the $m^7GTP$ cap was impaired in $Atg12^{KO}$ cells, there was not significant overlap in the mRNAs whose translation is affected by autophagy deletion compared with mRNAs whose translation is affected by eIF4A1 inhibitors[32] (Supplementary Fig. 6c). Notably, p62/SQSTM1 has been previously implicated in the sequestration of the E3 ligase KEAP1 away from its target substrate NRF2 in autophagy-deficient cells[33]. Based on our results, we propose a similar model in which the accumulated p62/SQSTM1 in $Atg12^{KO}$ cells sequesters eIF4A1 away from the translation initiation complex. We postulate that eIF4A1 availability is determined by p62/SQSTM1 levels, which contributes to autophagy-enabled translation, but this not the sole determinant of the sensitivity of Brca2 translation to autophagy inhibition.

**Autophagy impacts the availability of RBPs to bind to Brca2.** We investigated whether other RBPs that bind to the Brca2 5′

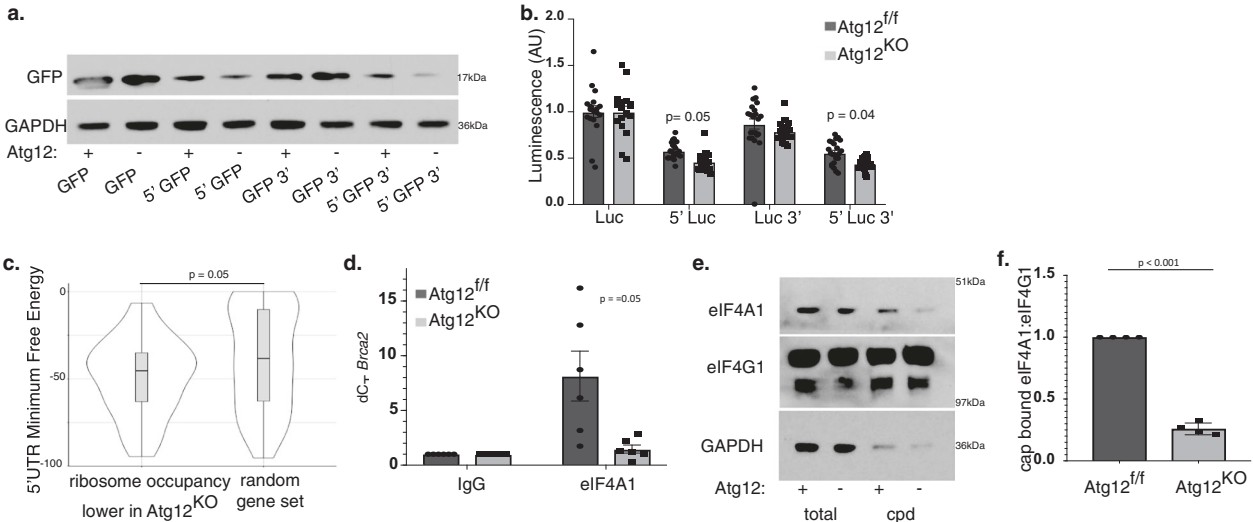

**Fig. 5 The 5′UTR of Brca2 determines translational sensitivity to autophagy due to structure complexity, requiring the helicase eIF4A1. a** *Atg12*[f/f] and *Atg12*[KO] MEFs were transfected with pcDNA3 expressing *Gfp*, *Gfp* preceded by the 5′UTR of *Brca2*, *Gfp* followed by the 3′UTR of *Brca2*, or *Gfp* flanked by both 5′ and 3′ UTRs of *Brca2*. Representative immunoblot for levels of GFP is shown, three independent biological replicates performed. **b** *Atg12*[f/f] and *Atg12*[KO] MEFs were transfected with pNL1.1 driving the expression of nano-luciferase, nano-luciferase preceded by the 5′UTR of *Brca2*, nano-luciferase followed by the 3′UTR of *Brca2*, or nano-luciferase flanked by both 5′ and 3′ UTRs of *Brca2*. Luciferase activity was measured by Nano-glo (Promega). Quantification shown (mean ± SEM, $n = 4$ independent biological replicates), statistics calculated by *t* test for biological replicates only. **c** Local minimum free energy (MFE) was predicted by RNALfold in the 5′UTRs from mRNAs with significantly lower than expected ribosome occupancy in *Atg12*[KO] MEFs compared with a randomly generated gene set. A violin plot with boxplot overlay of the MFEs is shown, $p = 0.05$ by Kolmogorov–Smirnov test. **d** Quantification (mean ± SD, $n = 6$ independent biological replicates) of the fold enrichment of Brca2 interaction with eIF4A1 over IgG control in *Atg12*[f/f] or *Atg12*[KO] MEFs by RNA immunoprecipitation. $p = 0.05$ by *t* test. **e** Protein lysate from *Atg12*[f/f] and *Atg12*[KO] MEFs was captured by cap pulldown, and total protein lysate and pulldown was immunoblotted as indicated. **f** Quantification (mean ± SD, $n = 4$ independent biological replicates) of eIF4A1 capture by cap pulldown relative to eIF4G1 capture. $p = 7.4E{-}08$ by *t* test.

UTR were affected by autophagy deficiency. The 5′UTR of Brca2 contains two predicted binding sites for the RBP MSI1, which represses the translation of p21 and Numb[34]. We therefore assayed MSI1 binding to Brca2 by RNA immunoprecipitation and observed increased MSI1 associated with Brca2 in the *Atg12*[KO] cells (Fig. 7a). *Atg12*[KO] MEFs in both fed and starved conditions demonstrated a modest, albeit not significant, accumulation of MSI1 of a similar pattern and magnitude that inversely correlated with BRCA2 decrease (Fig. 7b). MSI1 possesses four putative LC3 interacting domains (LIRs)[35] and we found that MSI1 interacts with multiple LC3/ATG8 orthologues (Fig. 7c), suggesting that MSI1 is selectively targeted by autophagy. To further define how MSI1 affected Brca2 translation, we depleted MSI1 in Atg12[f/f] and Atg12[KO] MEFs by shRNA (Fig. 7d), which reduced the steady state levels of BRCA2 compared with nontargeting control shRNA in both cell types ($p = 0.23$ by ANOVA) (Fig. 7e, f). Nonetheless, upon MS1 depletion, the reduction in BRCA2 protein levels in *Atg12*[KO] compared with *Atg12*[f/f] MEFs was significantly less pronounced (Fig. 7g).

Furthermore, the analysis of two published data sets of the autophagy interactome[36] and a comprehensive list of RNA-binding proteins[37] revealed that nearly one quarter of the autophagy interactors are RBPs (Supplementary Fig. 7a). We therefore questioned whether common RNA-binding motifs were more prevalent in the UTRs of significant RP hits. Using RBPDB[38], we analyzed the UTRs of the significant hits from our RP analysis compared with a randomly generated gene list of equivalent length, revealing that multiple RBP motif sequences located at both 5′ and 3′ UTRs of the significant RP hits were enriched above the control set (Supplementary Fig. 7b). We specifically identified changes in the number of motifs for the RNA-binding proteins YBX1 and MATR3 (Supplementary Fig. 7c) and corroborated that both of these RBPs interacted

with various LC3/ATG8 orthologues (Supplementary Fig. 7d). However, we did not observe their accumulation in *Atg12*-deleted cells, either in full media or HBSS starvation conditions, suggesting these RBPs did not undergo significant turnover via starvation-induced autophagy (Supplementary Fig. 7e). Based on our results, we postulate that Brca2 translation is partly controlled by the autophagic turnover of MSI1, and that the regulation of other mRNAs in an autophagy sensitive manner likely arises from the coordinate regulation of multiple LC3/ATG8-interacting RBPs, including eIF4A1, YBX1, and MATR3.

**Atg12[KO] cells accumulate DNA damage and centrosome defects**. We next dissected the functional consequences of reduced Brca2 translation in autophagy-deficient cells. BRCA2-deficient cells are impaired in homologous recombination and accumulate DNA damage[39]. We observed increased levels of DNA damage in *Atg12*[KO] MEFs (independent of Cre expression), evidenced by increased levels of γH2AX, a marker of double strand DNA damage, by immunofluorescence and immunoblotting, as well as increased puncta double positive for γH2AX and 53BP1 by immunofluorescence (Fig. 8a, b, Supplementary Fig. 8a). Similar increases in γH2AX were observed in *Atg5*[KO] MEFs and autophagy-deficient 293T cells (Supplementary Fig. 8b).

Previous work attributes increased DNA damage in autophagy-deficient mammalian cells to reactive oxygen species (ROS) produced from accumulated defective mitochondria[40,41]. However, we observed no significant differences in ROS levels, mitochondrial mass, or membrane potential between *Atg12*[f/f] and *Atg12*[KO] cells (Fig. 8c, Supplementary Fig. 8c). To determine the functional contributions of ROS versus BRCA2 levels to DNA damage accumulation in *Atg12*[KO] MEFs, we measured γH2AX levels in cells treated with the ROS scavenger N-acetyl cysteine (NAC), or ectopically expressing the human *Brca2* cDNA (Supplementary

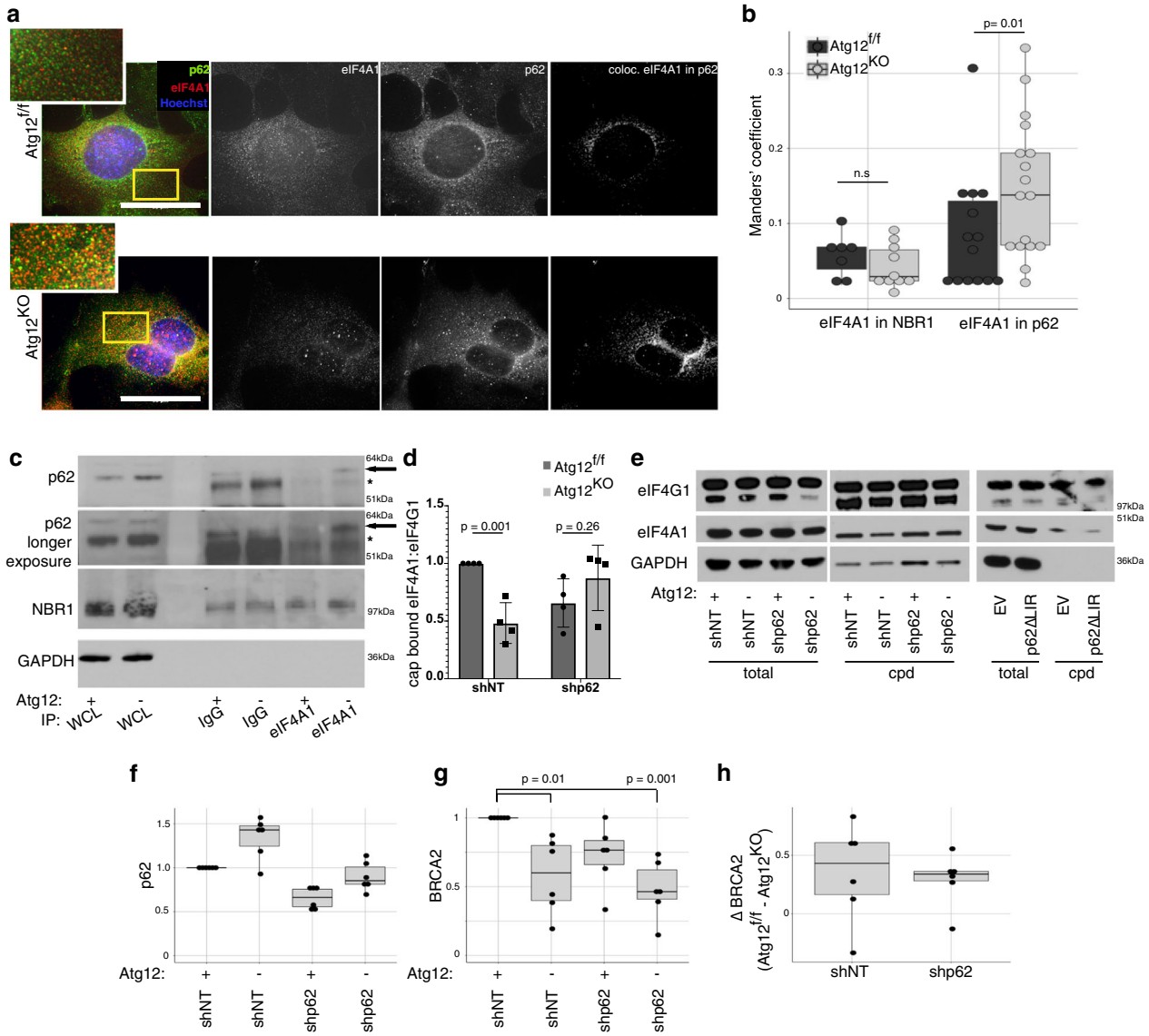

**Fig. 6 eIF4A1 is sequestered by accumulated p62 in autophagy-deficient cells. a** Representative immunofluorescence images for eIF4A1 (red in merged imaged) and p62/SQSTM1 (green in merged imaged) in *Atg12*^f/f and *Atg12*^KO MEFs, nuclei were counterstained with Hoechst (blue). Yellow box indicates region of inset panel in the top left corner. Far right panels show the points of colocalization (white) of eIF4A1 in p62/SQSTM1. Bars = 50 μm. **b** Manders' coefficient (percent of colocalization) of eIF4A1 in either NBR1 or p62/SQSTM1 in *Atg12*^f/f and *Atg12*^KO MEFs was calculated, and boxplot with dotplot overlay representing one field is shown (*n* = 3 biologically independent replicates). An outlier (Dixon test *p* = 0.03) was excluded from statistical analysis and the *p* values between *Atg12*^f/f and *Atg12*^KO were calculated by *t* test. **c** Representative immunoprecipitation of eIF4A1 and immunoblot for the autophagy cargo receptors p62/SQSTM1 and NBR1. Arrow indicates p62/SQSTM1, asterisk indicates immunoglobulin heavy chain. Immunoprecipitation was performed with three biologically independent replicates. **d** Protein lysate from *Atg12*^f/f and *Atg12*^KO MEFs that were knocked down for p62/SQSTM1 or treated with nontargeting shRNA was captured by cap pulldown, and the ratio of eIF4A1 to eIF4G1 was quantified (mean ± SD, *n* = 4 biologically independent replicates). **e** Protein lysate from *Atg12*^f/f and *Atg12*^KO MEFs that were knocked down for p62/SQSTM1 or treated with nontargeting shRNA or HEK293Ts that overexpress p62ΔLIR or empty vector control was captured by cap pulldown, and immunoblotted as indicated. **f–h** Protein lysate from *Atg12*^f/f or *Atg12*^KO MEFs stably infected with shRNA to p62/SQSTM1 or nontargeting control was collected. Relative quantification from immunoblots for **f** p62/SQSTM and **g** BRCA2, normalized to loading control are shown in the boxplot with dotplot overlay per biological replicate. *p* values were calculated using ANOVA with Tukey's post hoc test and significant values are reported. **h** The change in BRCA2 levels between *Atg12*^f/f and *Atg12*^KO in shNT-treated cells or shp62-treated cells is plotted, *p* = 0.76 between shNT and shp62 for ΔBRCA2, calculated by *t* test.

Fig. 8d). Whereas treatment with NAC had minimal effects on γH2AX levels in *Atg12*^KO cells, overexpression of *Brca2* decreased the levels of γH2AX in *Atg12*^KO cells (Fig. 8d) and in ATG-deleted HEK293T cells (Supplementary Fig. 8e). Overall, these results indicate that the impaired translation of Brca2 functionally contributes to DNA damage in autophagy-deficient cells.

We next treated *Atg12*^KO cells with the Poly ADP-ribose polymerase (PARP) inhibitors rucaparib, olaparib, and BMN to assess if reduced BRCA2 protein levels conferred sensitivity to inhibition of single-strand DNA damage repair, as previously observed in the context of BRCA2 genetic deficiency[42,43]. Indeed, *Atg12*^KO cells exhibited increased sensitivity to PARP inhibitors, evidenced by increased γH2AX and cleaved caspase 3 levels (Fig. 8e, Supplementary Fig. 8f), as well as impaired colony-replating efficiency following PARP inhibitor treatment (Fig. 8f).

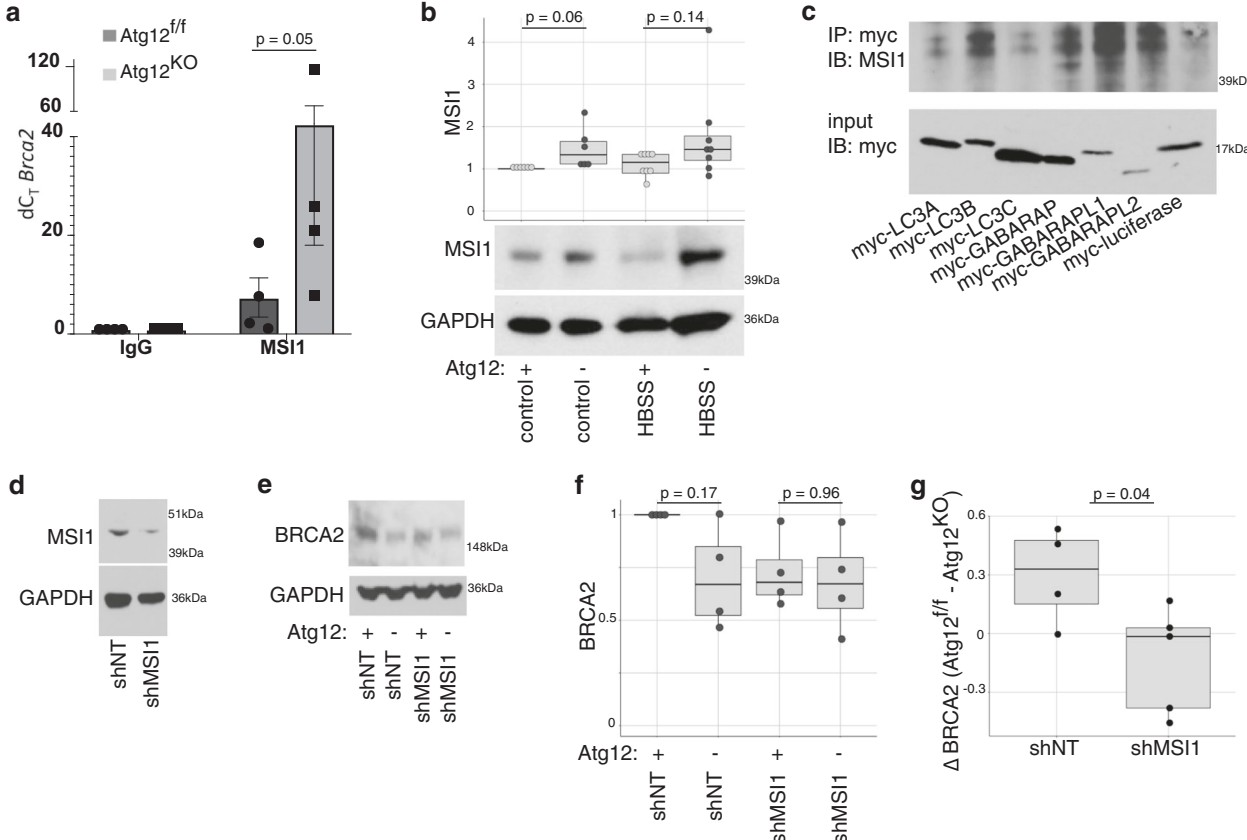

**Fig. 7 Increased MSI1 in autophagy-deficient cells impairs Brca2 translation efficiency. a** Quantification (mean ± SD, $n = 4$ biologically independent replicates) of the fold enrichment of Brca2 interaction with MSI1 over IgG control in *Atg12*^f/f or *Atg12*^KO MEFs by RNA immunoprecipitation. An outlier (Dixon test $p = 0.007$) was excluded from statistical analysis, and the $p$ value by $t$ test is shown. **b** Boxplot, with dotplot overlay for each biological replicate, of relative MSI1 protein levels normalized to loading control and representative immunoblots from autophagy-inhibited MEFs, assayed by immunoblotting. Statistical analysis was performed by $t$ test. **c** Representative immunoblot of immunoprecipitation of myc-tagged overexpressed LC3 family members interaction with MSI1 in HEK293T cells. Immunoprecipitation was performed with three biologically independent replicates. **d** Protein lysate was collected from *Atg12*^f/f MEFs treated with shRNA to Msi1, and immunoblotted as indicated. Knockdown of ~50% was consistent among three independent biological replicates. **e–g** Protein lysate was collected from *Atg12*^f/f MEFs that were stably knocked down for MSI1 and subsequently treated with 4OHT or control, and assayed by immunoblotting. **e** Representative immunoblot is shown. **f** Quantification of relative BRCA2 protein levels normalized to loading control, are plotted as a boxplot with dotplot overlay for each independent biological replicate, $p$ values by ANOVA with Tukey's post hoc test. **g** Difference in BRCA2 steady state protein levels (ΔBRCA) between *Atg12*^f/f and *Atg12*^KO MEFs following MS1 depletion, $p$ value by $t$ test.

BRCA2 contributes to clustering of mother and daughter centrosomes following duplication[44]. We observed similar impairments in centrosome clustering in *Atg12*^KO cells. The distance between the two centrosomes in non-mitotic cells was increased (Fig. 8g, h) and there was a significant increase in percentage of cells with more than two centrosomes in *Atg12*^KO cells compared with wild-type controls (Fig. 8g, i). These defects in centrosome organization may further exacerbate DNA damage in *Atg12*^KO cells[45]. In addition to Brca2, our ribosome profiling analysis identified other mRNAs involved in centrosome function, including Haus3 and Cntln, that exhibited reduced ribosome occupancy in *Atg12*^KO cells (Supplementary Data 2), suggesting that autophagy-dependent translation of multiple mRNAs may functionally contribute to centrosome organization.

**Autophagy deletion in vivo reduces BRCA2.** We assessed the effects of autophagy ablation on BRCA2 protein levels in vivo following systemic acute genetic deletion of *Atg12* in adult mice. At 6 weeks of age, *Atg12*^f/f *Cag-Cre*^ER mice were subject to treatment with tamoxifen or vehicle control for 5 consecutive days (Fig. 9a)[46]. Loss of ATG12 correlated with accumulation of the autophagy substrate p62/SQSTM1 and the absence of LC3-II

at 2 weeks following tamoxifen administration (Fig. 9b). *Atg12*^KO animals survived for 10 weeks following the acute loss of autophagy. Similar to acute systemic genetic deletion of *Atg7* in adult mice[47], systemically deleted *Atg12* mice were smaller and failed to gain weight following deletion (Fig. 9c, d). Immunoblotting revealed decreased BRCA2 protein levels in the cerebral cortex of *Atg12*^KO mice compared with autophagy competent *Atg12*^f/f controls, and a non-significant decrease in the kidney (Fig. 9e, f). This correlated with increased levels of DNA damage, evidenced by a twofold increase in γH2AX-positive nuclei in the kidney, cerebral cortex, and small intestine of *Atg12*^KO mice (Fig. 9g, h). These in vivo findings are consistent with our in vitro results that an autophagy pathway supports the production of BRCA2.

## Discussion
Using ribosome profiling, we have uncovered that autophagy is functionally required for the efficient translation of specific proteins, and in particular BRCA2, in mammalian cells. Strikingly, our results show important differences in the starvation response between mammalian cells and *Saccharomyces cerevisiae*, which rely heavily on autophagy to maintain amino acid availability and

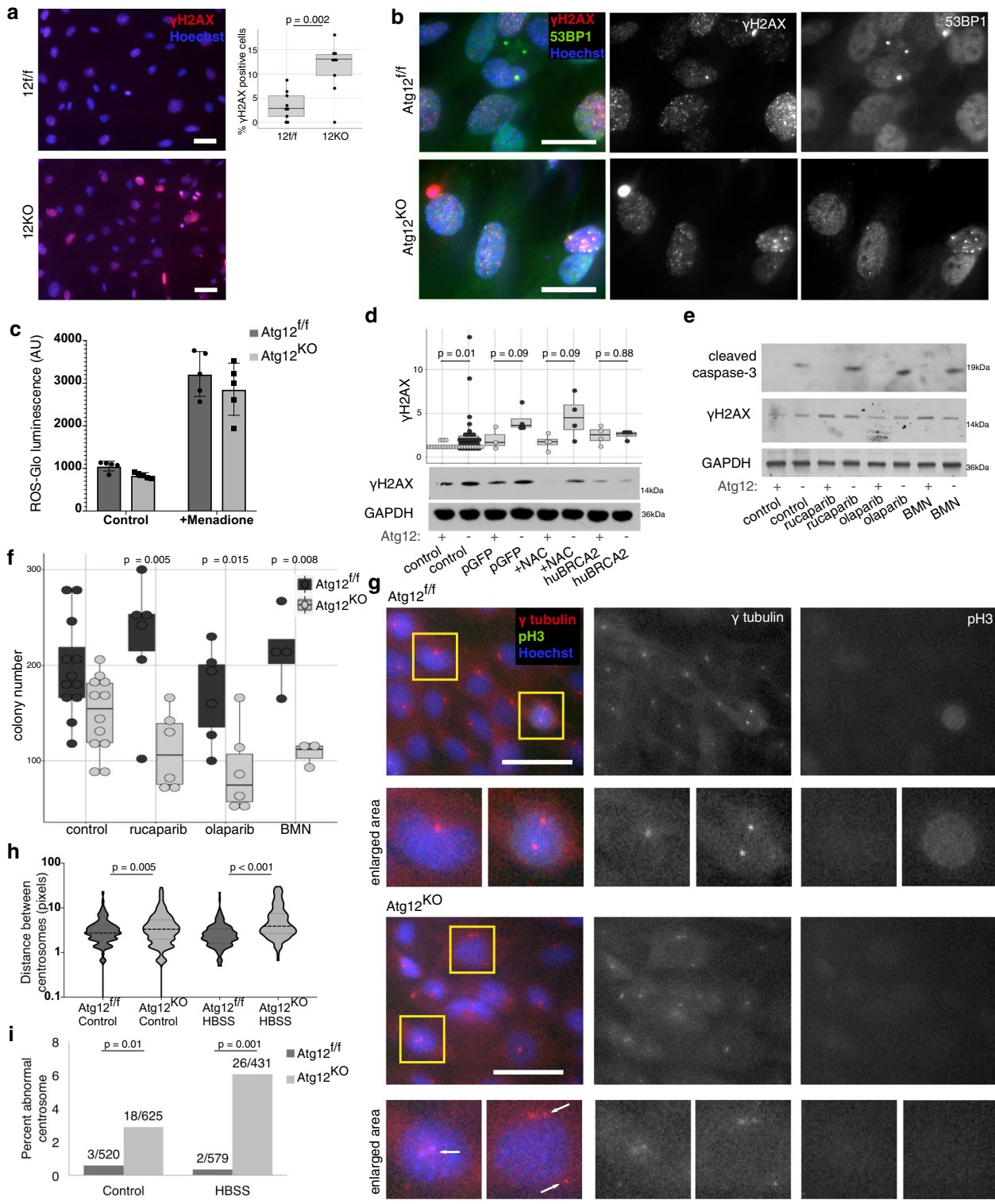

protein synthesis during starvation[7,48]. In contrast, global protein synthesis and the availability of intracellular amino acids remains largely intact in mammalian cells following autophagy ablation, including cells undergoing short-term starvation, suggesting that other proteolytic pathways, such as direct delivery of ER and plasma membrane components to the lysosome[49,50], likely compensate to maintain amino acid levels in response to acute stress. Because of both the specific regulation of a small subset of mRNAs, and the minimal changes observed in amino acid availability in autophagy-deficient cells, we propose that the effects of autophagy deficiency on the translational control are unlikely due to altered cellular metabolism. In contrast to our work, previous studies in several cancer models demonstrate that autophagy supports local and serum amino acids levels necessary to fuel tumor growth[51–54]. These dependencies may be specific to tumor cells, which are highly reliant on glutamine levels, or indirectly arise from the autophagy-regulated secretion of specific factors in certain cells and tissues, such as arginase or alanine.

**Fig. 8 Decreased BRCA2 levels in Atg12-deleted cells result in increased DNA damage and centrosome abnormalities. a** Immunofluorescence for γH2AX (red) in *Atg12*$^{f/f}$ and *Atg12*$^{KO}$ MEFs; nuclei are counterstained with Hoechst (blue). Bar = 50 μm. Percent of γH2AX-positive cells was quantified in *Atg12*$^{f/f}$ and *Atg12*$^{KO}$ MEFs; three random fields were counted over three independent biological replicates, $p = 0.002$ by $t$ test. **b** Immunofluorescence for γH2AX (red) and 53BP1 (green) in *Atg12*$^{f/f}$ and *Atg12*$^{KO}$ MEF; nuclei are counterstained with Hoechst (blue). Bar = 50 μm. Three biologically independent replicates were performed. **c** ROS-glo assay (Promega) in *Atg12*$^{f/f}$ and *Atg12*$^{KO}$ MEFs treated with vehicle control or menadione (50 μM for 2 h, positive control) (mean ± SEM, $n = 2$ biologically independent replicates). **d** Protein lysate was collected from *Atg12*$^{f/f}$ and *Atg12*$^{KO}$ MEFs treated with vehicle control or NAC (5 mM for 8 h), or ectopically overexpressing either GFP (pGFP) or human BRCA2 (huBRCA2). A representative immunoblot for γH2AX is shown, as well as boxplots with independent biological replicates, for γH2AX levels normalized to loading control. Statistical analysis was performed by $t$ test. **e** Protein lysate was collected from *Atg12*$^{f/f}$ and *Atg12*$^{KO}$ MEFs treated for 16 h with vehicle control, rucaparib (100 nM), olaparib (100 nM), or BMN (2 nM), and immunoblotted as shown. Three independent biological replicates were performed. **f** A clonogenic replating assay was performed on *Atg12*$^{f/f}$ or *Atg12*$^{KO}$ MEFs treated for 16 h with vehicle control, rucaparib (100 nM), olaparib (100 nM), or BMN (2 nM). Colony number is shown as a boxplot including biological replicates, $p$ value by $t$ test, **g** Immunofluorescence of centrosomes (γ-tubulin, red) and mitotic cells (pH3, green), nuclei counterstained by Hoechst (blue), in *Atg12*$^{f/f}$ and *Atg12*$^{KO}$ MEFs. Yellow box indicates magnified region below. White arrows indicate non-mitotic cells with multiple centrosomes (3+) or non-clustered centrosomes. Bar = 100 μm. **h** Quantification (mean ± SEM, $n = 3$ biologically independent replicates) of distance between mother and daughter centrosomes measured on immunofluorescence of γ-tubulin in non-pH3 positive cells, $p$ value by $t$ test. **i** Quantification of *Atg12*$^{f/f}$ and *Atg12*$^{KO}$ MEFs with abnormal numbers (3+) of centrosomes from immunofluorescence images ($n = 4$ biologically independent replicates). $p$ values by $t$ test on logit transformed percent per replicate. Fraction above the bar plots indicates number of cells with 3+ centrosomes out of number of cells enumerated.

Importantly, we demonstrate that basal autophagy enables the efficient translation of a cohort of mRNAs by regulating the availability of certain RBPs. In particular, we show that autophagy enables the efficient production of the DNA damage repair protein BRCA2. As a result, when autophagy is inhibited, reduced levels of BRCA2 lead to increases in DNA damage and centrosome defects. We propose multiple indirect mechanisms of translational control, including the ability of autophagy to modulate the interaction of MSI1 with the 5′UTR of Brca2, which impairs efficient translation and production of BRCA2. In addition to these autophagy-dependent effects on the 5′UTR of Brca2, we recognize that a broader repertoire of translation control mechanisms are impacted by autophagy, including the modulation of additional LC3/ATG8-binding RBPs, such as eIF4A1, MATR3, and YBX1. Moreover, although we found no role for autophagy in controlling cap versus HCV or CrPV IRES translation initiation, we cannot rule out that autophagy may regulate translation from IRES-like or IRES motifs distinct from the viral motif driven reporter systems we employed. Notably, we observed a trending increase in the levels of p-eIF2α in autophagy-deficient cells, which may tune attenuated translation in response to stresses other than nutrient starvation. Overall, identifying the diverse array of molecular mechanisms by which autophagy impacts mRNA translation, both directly and indirectly, remains an important area for further study.

Autophagy enhanced Brca2 translation may have particular relevance for human health and disease. We found in vivo reductions in BRCA2 protein levels and increases in DNA damage, including a twofold reduction in BRCA2 levels and a threefold increase in γH2AX levels in the kidney, in *Atg12*-deleted mice. Polycystic kidney disease has been linked independently to both defects in autophagy and defects in centrosome organization that disrupt primary cilia formation[55–57]. Our results broach centrosome disorganization as a potential mechanism by which defective autophagy contributes to this disease phenotype. Furthermore, because intestinal stem cells and hematopoietic stem cells are highly dependent on autophagy to maintain genome integrity[58–60], our data suggest a previously unrecognized mechanism by which autophagy maintains the genome in stem cells. While ROS has been primarily implicated as the DNA damaging driver in autophagy-deficient intestinal stem cells, our results identify reduced Brca2 translation as an aggravating factor contributing to DNA damage in autophagy-deficient cells. With regard to cancer, one can speculate autophagy mitigates genomic damage by enabling the translation of Brca2, thereby suppressing early tumorigenesis. In support, a

polymorphism in the 5′UTR of Brca2 that decreases the secondary structure and promotes translation is protective against breast cancer in patients[61]. Because autophagy inhibitors are being tested as adjuvant chemotherapies[62], further defining the effects of autophagy on protein synthesis in cancer cells will help refine the proper contexts to effectively employ such strategies. Overall, our findings illuminate roles for autophagy in directing the protein synthesis landscape in mammalian cells, which maintains genome integrity.

## Methods

### Experimental models and subject details

*Mouse maintenance.* Compound transgenic C57Bl/6 mice harboring *Atg12*$^{f/f}$ and *Cag-Cre*$^{ER}$ were generated by cross-breeding of *Atg12*$^{f/f}$[10] and *Cag-Cre*$^{ER}$ animals (obtained via the UCSF mouse database). Offspring were genotyped with polymerase chain reaction (PCR) primers (msAtg12$^{f/f}$ primers: FRT h/h sense 1: ATG TGA ATC AGT CCT TTG CCC; FRT-FRT as 2: ACT CTG AAG GCG TTC ACG GC; WT-FRT as 2: CTC TGA AGG CGT TCA CAA CA. Cag-CreER primers: forward: GCCTGCATTACCGGTCGATGC, reverse: CAGGGTGTTATAAGC AATCCC; msAtg12-null allele: forward: CACCCTGCTTTTACGAAGCCCA, reverse: ACTCTGAAGGCGTTCACGGC). At 6 weeks of age, animals of indicated genotypes received either Tamoxifen (0.2 mg/gram mouse; Sigma-Aldrich T5648-1G) or vehicle (peanut oil) via oral gavage for 5 consecutive days. At 10 weeks after the first tamoxifen treatment, animals were sacrificed and tissues were collected for biochemical and histological analysis. Both male and female animals were used in roughly equal numbers for all experiments. All experimental procedures and treatments were conducted in compliance with UCSF Institutional Animal Care and Use Committee guidelines.

*Isolation of mouse embryonic fibroblasts.* Mouse embryonic fibroblasts were generated from E13.5 mice described above following the protocol from Robertson[63]. Briefly, embryos were collected, heart, liver, and head were removed and fibroblasts were minced, digested in trypsin for 30 min at 37 °C and plated in DMEM with 10% serum and Pen/Strep. Cells were genotyped and Atg12$^{f/f}$;Cag-Cre$^{ER+}$ cells were immortalized by infection with SV40 large T antigen (SV40 1: pBSSVD2005, Addgene #21826, deposited by David Ron). Cells were plasmocin treated prior to use (Invivogen, ant-mpt). Following immortalization and plasmocin treatment, cells were maintained in DMEM 1x (Gibco) supplemented with 10% FBS (Atlas).

*Genetic deletion of MEFs.* Cells were treated with 2 μM 4-hydroxy-tamoxifen (4OHT; Sigma-Aldrich H7904) or vehicle (100% ethanol) for 3 consecutive days. Genetic recombination was achieved following 2 days of 4OHT treatment, and confirmed by PCR.

*Additional tissue culture cells.* N. Mizushima (University of Tokyo, Japan) provided *Atg5*$^{+/+}$, *Atg5*$^{-/-}$, *Atg7*$^{+/+}$, and *Atg7*$^{-/-}$ MEFs and M. Komatsu (Tokyo Metropolitan Institute, Japan) provided *Atg3*$^{+/+}$ and *Atg3*$^{-/-}$ MEFs. *Atg12*$^{+/+}$ and *Atg12*$^{-/-}$ MEFs were originally generated in Malhotra et al.[10]. HEK293Ts were cultured in DMEM 1x (Gibco) supplemented with 10% FBS (Atlas) and Pen/Strep. HEK293Ts were purchased from ATCC (CRL-3216). HEK293T knockout cell lines lacking ATG7, ATG12, or ATG14 were generated by CRISPR/Cas9. Human guide sequences (scramble: gcactaccagagctaactca; huAtg12: CCGTCTTCCGCTGCAGT

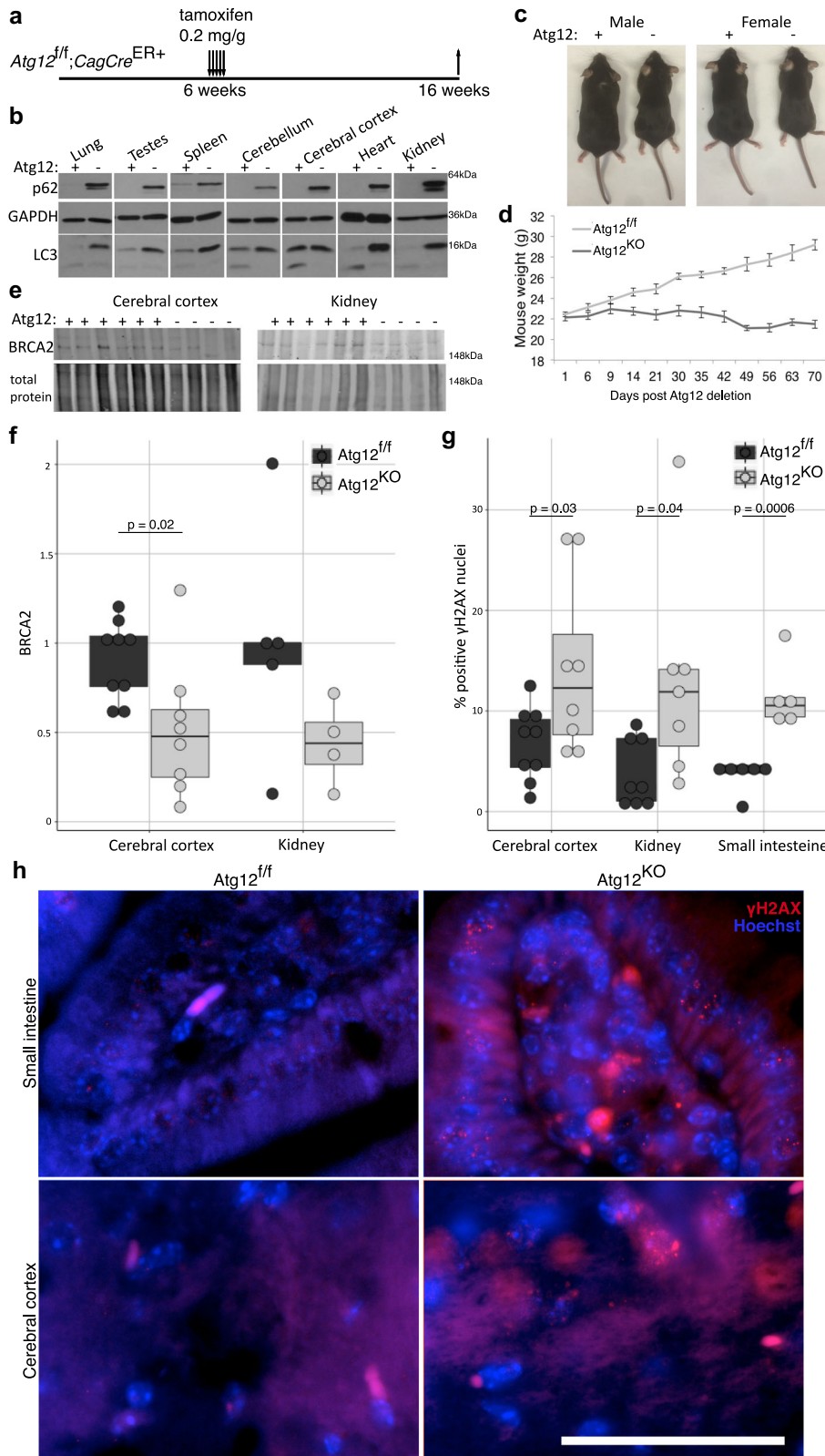

TTC; huAtg7: ACACACTCGAGTCTTTCAAG; huAtg14: CTACTTCGACGGCC GCGACC) were ligated into pSpCas9(BB)-2A-Puro (PX459) plasmid using the BbsI site. HEK293T cells were transfected with plasmid DNA using Lipofectamine 3000. Cells were selected 48–72 h post-transfection with 1 mg/ml puromycin for 48 h. Polyclonal populations were collected for Surveyor analysis (IDT, 706020) and were sorted into single-cell populations by limiting dilution at 1.5 cells/well per 96-well plate. Monoclonal wells were identified, expanded, and analyzed. For DNA analysis, genomic DNA samples were prepared using QuickExtract (Epicentre).

The PCR products were column purified and analyzed with Surveyor Mutation Detection Kit (IDT). For genotyping of single-sorted cells, PCR amplified products encompassing the edited region were cloned into pCR™4-TOPO® TA vector using the TOPO-TA cloning kit (Thermo Fisher #450030) and sequence verified (primers for genotyping CRISPR deleted HEK293T cells: Atg12: forward: AGCCGG GAACACCAAGTTT, reverse: GTGGCAGCCAAGTATCAGGC; Atg7: forward: TGGGGGACAGTAGAACAGCA, reverse: CCTGGATGTCCTCTCCCTGA; Atg14: forward: AAAATCCCACGTGACTGGCT, reverse: AATGGCAGCA

**Fig. 9 Atg12 deletion in vivo leads to reduced BRCA2 and increased DNA damage. a** Diagram of *Atg12*$^{f/f}$;*Cag-Cre*$^{ER+}$ mouse treatment and tissue collection. **b** Protein lysate was collected from tissues two weeks following vehicle or 0.2 mg/g tamoxifen treatment and immunoblotted for markers of autophagic flux (p62/SQSTM1, LC3). **c** Representative images of male and female *Atg12*$^{f/f}$ and *Atg12*$^{KO}$ littermates. **d** Body weights of male mice following *Atg12* deletion (*Atg12*$^{f/f}$ *n* = 17, *Atg12*$^{KO}$ *n* = 14). **e** Protein lysate was collected from mouse tissues 10 weeks following vehicle or tamoxifen treatment, and immunoblotted for BRCA2. **f** Boxplot with dotplot overlay for biological replicates of BRCA2 protein levels, normalized to total protein levels, assayed by immunoblotting. *p* = 0.02 by *t* test. **g** Boxplot with dotplot overlay for biological replicates of percent of γH2AX-positive nuclei by immunofluorescence, counted over four randomly selected fields of stained tissue per a minimum of four mice. *p* value calculated by *t* test. **h** Representative immunofluorescence for γH2AX (red) in mouse tissues from the cerebral cortex and small intestine, with nuclei counterstained by Hoechst (blue). Bar = 50 μm.

ACGGGAAAAC). Sequencing is available upon request. Cells were routinely tested for mycoplasma contamination and authenticated using short-tandem repeat (STR) profiling for human cell lines and immunoblotting for Atg expression in mouse cells.

*In vitro drug treatments.* Cells were treated with the following drugs at concentrations and times as indicated in the figure legends: Cycloheximide (Sigma-Aldrich, C7698), PP242 (Tocris 4257), Thapsigargin (Cayman Chemical Company 10522), Bafilomycin A (Sigma-Aldrich, B1793), Rucaparib (Selleck Chemicals, S1098), Olaparib (Cellagen Technologies, C2228-5s) BMN637 was a gift from Alan Ashworth[43], is commercially available from Selleck Chemicals (S7048), menadione (Sigma-Aldrich, M5625), N-acetyl cysteine (Sigma-Aldrich, A7250).

*Stable RNA interference.* pLKO.1 blasticidin or pLKO.1 puromycin lentiviral plasmids with nontargeting shRNA, which targets no known mammalian genes (SHC-002, CAA CAA GAT GAA GAG CAC CAA), or shRNA against mouse Atg7 (TRC N0000092163, CCA GCT CTG AAC TCA ATA ATA), or mouse Msi1 (TRCN000098550, CCGG CCTGTTCAGACCTTGTCTCTT CTCGAG AAGAG ACAAGGTCTGAACAGG TTTTTG) were purchased from Sigma-Aldrich. shRNA lentivirus was prepared by cotransfecting HEK293T cells with packaging and envelope vectors and pLKO.1 shRNA expression plasmids. Virus was collected 48 h after transfection, filtered through a 0.45-μm filter, and stored at −80 °C. Cells were seeded in six-well dishes and infected for generation of stable cell lines. Stable pools of knockdown cells were obtained by selecting with 2 ng/ml blasticidin or 1–2 μg/ml puromycin for 48 h.

*Ribosome profiling.* Ribosome profiling experiments were performed using the ARTseq Ribosome profiling kit (Epicentre RPHMR12126), with RNA extraction by Trizol LS (Ambion), rRNA depletion via RiboZero Gold (Epicentre MRZG126), and quality and quantity of small RNA and DNA assayed using Agilent High Sensitivity Small RNA kit and DNA kit, respectively (Agilent 5067-1548, 5067-4626). Sequencing was performed at the UCSF sequencing core on Illumina HiSeq2000, and analysis of reads was performed using Babel[20]. Cycloheximide was made fresh to 50 mg/ml in ethanol for each experiment, used at a concentration of 100 μg/ml (Sigma-Aldrich C7698).

*Immunoblotting.* For immunoblot analysis, 200,000–300,000 cells were lysed in RIPA buffer (1% Triton X-100, 1% sodium deoxycholate, 0.1% SDS, 25 mM Tris, pH 7.6, and 150 mM NaCl) plus protease inhibitor cocktail (Sigma-Aldrich), 10 mM NaF, 10 mM β-glycerophosphate, 1 mM Na$_3$VO$_4$, 10 nM calyculin A, 0.5 mM PMSF, 10 μg/ml E64d, and 10 μg/ml pepstatin A. Lysates were freeze-thawed at −20 °C, cleared by centrifugation for 30 min at 4 °C, protein content was quantified by BCA assay and equal amounts were boiled in sample buffer, resolved by SDS-PAGE, and transferred to polyvinylidene fluoride membrane. Membranes were blocked for 1 h in 5% milk or 5% BSA in PBS with 0.1% Tween 20, incubated in primary antibody (1:1000 unless otherwise specified) overnight at 4 °C, washed, incubated for 1 h at RT with HRP-conjugated goat secondary antibodies (1:5000; Jackson ImmunoResearch Laboratories; streptavidin HRP conjugate Thermo Scientific Pierce 21130), washed, and visualized via enhanced chemiluminescence (Thermo Fisher Scientific). Antibodies used for immunoblotting are anti-BRCA2 rabbit pAb from Bioss USA bs1210R (1:300); anti-p62 guinea pig pAb from Progen GP-62-C; anti-LC3 mouse pAb, Fung et al.[64], now commercially available from EMD Millipore ABC232; anti-phospho-S6 S240/244 rabbit mAb Cell Signaling #2215; anti-S6 rabbit mAb Cell Signaling #2217; anti-α-tubulin rabbit mAb Cell Signaling #2125BC; anti-phospho-4EBP1 S65 rabbit pAb Cell Signaling #9451; anti-4EBP1 rabbit pAb Cell Signaling #9452; anti-GAPDH mouse mAb Millipore MAB374; anti-phospho-eIF2α S51 rabbit pAb Cell Signaling #9721; anti-eIF2α S51 rabbit pAb Cell Signaling #9722; anti-eIF4G1 rabbit mAb Cell Signaling #2469; anti-eIF4G2 rabbit pAb Cell Signaling #2182; anti-eIF4E rabbit pAb Cell Signaling #9742; anti-eIF4E2 rabbit pAb Fierce P A5-11798; anti-Atg12 (mouse specific) rabbit pAb Cell Signaling #2011BC; anti-Atg5 rabbit pAb Novus Biologicals NB110-53818; anti-Atg7 rabbit pAb Cell Signaling #2631; anti-Atf4 rabbit mAb Cell Signaling #11815; anti-Cdc25a rabbit pAb Cell Signaling #3652; anti-Cpt2 Abcam ab71435; anti-Pfkfb3 rabbit pAb ABclonal Biotech A6945; anti-eEF2 rabbit pAb Cell Signaling #2332; anti-Mcl-1 rabbit pAb Rockland 800-401-394S; anti-Irf7 rabbit mAb Abcam ab109255; anti-γH2AX S139 mouse mAb Upstate #05-636; anti-cleaved Caspase 3 (Asp175) Rabbit pAb Cell Signaling #9661; anti-eIF4A1

rabbit pAb Cell Signaling #2490; anti-MSI1 rabbit pAb EMD Millipore AB5977 (1:500); anti-GFP mouse mAb Neuromab N86/8; anti-puromycin mouse mAb Kerafast EQ0001.

*Metabolic labeling.* In total, 200,000 cells were grown in 6-well plates and incubated for 1.5 h DMEM lacking methionine, DMEM lacking methionine and glucose, DMEM lacking methionine and glutamine (UCSF cell culture facility) or HBSS (Gibco), upon which 30 μCi of exogenous 35-S L-Methionine (Perkin Elmer NEG009A001MC) was added for the last 30 min. Cells were washed once in ice-cold PBS, lysed in RIPA buffer, protein content was quantified by BCA assay and equal total protein per sample was run on SDS-PAGE gels and transferred to PVDF as described above. For puromycin labeling, the cells were maintained in normal growth media, media lacking glutamine or glucose, or HBSS, and puromycin was added concentration of 10 μM for 30 min. Following transfer onto PVDF, the membrane was immunoblotted using an anti-puromycin antibody.

*AHA labeling.* Azidohomoalanine (Thermo Fisher Scientific C10102) was added to methionine-free DMEM at a final concentration of 40 μM and left to incorporate in HEK293Ts for 8 h. Subsequently, cell lysate was collected for immunoprecipitation. Following immunoprecipitation, the azide-alkyne conjugation reaction was performed using Diazo Biotin alkyne (Jena Biosciences CLK-1042-10) and the ClickIT kit (Thermo Fisher Scientific C10276) in a quarter volume, according to manufacturer's instructions.

*Immunoprecipitation, RNA immunoprecipitation, cap-pulldown assays.* Antibodies used for immunoprecipitation are anti-BRCA2 rabbit pAb Abcam ab123491, anti-MSI1 EMD Millipore AB5977, anti-eIF4A1 Cell Signaling #2490, anti-c-Myc clone 9E10 Sigma-Aldrich M5546, and normal rabbit IgG Santa Cruz Biotechnology sc2027. Cells were lysed in the following buffers: immunoprecipitation (IP) buffer: 25 mM Tris HCl pH 7.4, 150 mM NaCl, 1% NP-40, 5% glycerol, 1 mM EDTA, 1 mM EGTA, 1 mM β-glycerophosphate, 10 mM NaF, 2.5 mM NaP$_2$O$_7$, 1 μM sodium orthovanadate, plus protease inhibitor cocktail. RNA immunoprecipitation (RIP) buffer: 200 mM NaCl, 25 mM Tris HCl pH 7.4, 5 mM EDTA, 5% glycerol, 1 mM DTT, 1% NP-40, plus protease inhibitor cocktail and RNase inhibitors. Cap-pulldown (CPD) buffer: 10 mM Tris HCl pH 7.6, 140 mM KCl, 4 mM MgCl$_2$, 1 mM DTT, 1 mM EDTA, 1% NP-40, 1 mM PMSF, protease inhibitor cocktail, 0.2 mM sodium orthovanadate.

For immunoprecipitation (IP) and RNA immunoprecipitation (RIP), lysates were precleared with protein A/G (Santa Cruz sc-2003) and incubated on a rotating shaker overnight at 4 °C with protein A/G plus antibody. Four micrograms of antibody was added to 2-mg cell lysate. For IP, beads were washed four times with IP buffer, eluted in 3x sample buffer and analyzed by immunoblotting. For RIP, washed beads were split into a protein fraction and an RNA fraction. The protein fraction was subject to IP as described above. The RNA fraction was extracted with Trizol LS and bound RNA was analyzed by qPCR. For cap-pulldown experiments (CPD), cells were lysed in buffer listed below. About 25–50 μl of m7-GTP beads (γ-amino-phenyl-m7 GTP (C10-spacer) Jena Biosciences AC155S) were added to 250–500 μg protein at 1 μg/μl and incubated overnight. Beads were washed four times in CPD buffer, eluted in 3x sample buffer and analyzed by immunoblotting.

*Immunofluorescence.* In total, 20,000 cells were grown on fibronectin-coated (10 μg/ml in PBS) coverslips. Cells were fixed with 4% PFA for 5 min at RT, permeabilized with 0.5% Triton X-100 in PBS, rinsed with PBS-glycine, and blocked overnight at 4 °C in blocking buffer (10% goat serum and 0.2% Triton X-100 in PBS). Primary antibodies used were anti-γH2AX S139 Cell Signaling #9718S (1:200); anti-53BP1 rabbit pAb Abcam ab21083 (1:200); anti-γ tubulin mouse mAb Sigma T5326 (1:500); anti-phospho-Histone H3 S10 rabbit pAb Cell Signaling #9701 (1:500); anti-p62 guinea pig pAb Progen GP-62-C (1:500); anti-eIF4A1 rabbit pAb Cell signaling #2490 (1:200); anti-NBR1 (4BR) mouse mAb Santa Cruz Biotechnology sc1030380 (1:200). Cells were incubated with primary antibodies for 40 min at RT, washed, incubated with Alexa-Fluor 488 or 594 goat secondary antibodies (1:200; Life Technologies) for 40 min at RT, washed, nuclei were stained using Hoescht (Intergen S7304-5) and mounted using Prolong Gold Anti-Fade mounting medium (Life Technologies). Mitochondria analysis was performed using 500 nM MitoTracker Red CMX-Ros for 15 min (Thermo Fisher Scientific M7512) and 10 ng/ml DiOC6(3) (3,3′-dihexyloxacarbocyanine iodide) for 5 min (Thermo Fisher Scientific D273).

Tissues were paraffin embedded and sectioned by the UCSF Helen Diller Family Cancer Center mouse pathology core. Deparaffinization in xylene followed by antigen retrieval per manufacturer's instructions (Dako) was performed prior to immunofluorescence staining.

Epifluorescence images were obtained at ambient temperature using an Axiovert 200 microscope (Carl Zeiss) with a ×10 (NA, 0.25) or ×20 (NA, 0.4) objective, Spot RT camera (Diagnostic Instruments). High-magnification images were taken using the DeltaVision deconvolution microscope (Applied Precision) with a 60 1.42 NA Plan Apo objective (Olympus).

*Image analysis.* Immunoblot band intensity quantification was performed using ImageJ software. Immunofluorescence colocalization was performed using ImageJ software (JACoP plugin).

*Molecular cloning.* GFP and luciferase reporters were created by cloning the UTR sequences of BRCA2 into pcDNA3.EGFP plasmid (Addgene #13031, deposited by Doug Golenbock) and pNL1.1nano-luciferase plasmid (Promega N1001), using the primers described below. The ~500 bp 5′UTR of Brca2 that encompasses the region present in the shorter isoform of the Brca2 5′UTR was cloned using Gibson cloning, and the 3′UTR was cloned between restriction enzyme sites XhoI and XbaI. Primers for msBrca2 5′UTR cloning: forward: CGACTCACTATAGGGA GACCCAAGCTTGGTACCGGGCTTTTCGCGGGAGCGGG, reverse: TTTTTGT TCCATGGTAGATCCGAGTCTGGTACCTTCTCTACAGTATTTCTCCGA TGC TCG. Primers for msBrca2 3′UTR cloning: forward: GATCCTCGAGCCTCCCG GTTTGTAAGATGTGTACAGTTC, reverse: GATCTCTAGATTACAGCTGAA GTTCAGTGAGAGCATCCAC.

Human LC3B (NM_022818.4), LC3A (NM_032514.3), LC3C (NM_001004343.2), GABARAP (NM_007278.1), GABARAPL1 (NM_031412.2), and GABARAPL2 (NM_007285.6) were subcloned from mRNA isolated from human cell lines, reverse transcribed using AccuScript High Fidelity Reverse Transcriptase (Agilent), and cDNA amplified using PfuUltra II Hot Start DNA polymerase and gene specific primers (human LC3B: Fwd: agtcggatccatgccgtcgga gaagacct, Rev: gactctcgagttacactgacaatt tcatcccg; human LC3A: Fwd: agtcggatccatgc cctcagaccggcct, Rev: gactctcgagtcagaagccgaaggtttcct; human LC3C: Fwd: agtcggatc catgccgcctccacagaaaat, Rev: gactctcgagctagagaggattgcagggtc; human GABARAP: Fwd: agtcggatccatgaagttcgtgtacaaagaaga, Rev: gactctcgagttaaagaccgtagacactttc; human GABARAPL1: Fwd: agtcggatccatgaagttccagtacaaggac, Rev: gactctcgagtcatttc ccatagacactctc; human GABARAPL2: Fwd: agtcagatctatgaagtggatgttcaaggag, Rev: gactctcgagtcagaagccaaaagtgttctc). The cDNAs were subcloned into pcDNA3 between the BamHI and XhoI or EcoRI and XhoI restriction sites downstream of an N-terminal myc-tag or 3xFlag-tag. All constructs were verified by sequencing.

*Plasmid overexpression.* MEFs were transfected using the Amaxa Nucleofector device (Lonza), program T-020, MEF 1 nucleofector kit (Lonza VPD-1004), and 2 μg DNA, according to manufacturer's instructions. For all transfections, efficiency was monitored by qPCR.

*qPCR primers and reagents.* qPCR was performed using the StepOne Plus Real-Time PCR machine and Brilliant II SYBR green QRT-PCR 1-Step master mix (Agilent Technologies 600825). msBrca2 primers: forward: CTTACCGAGCATCG GAGAAA, reverse: CCGTGGGGCTTATACTCAGA; Gfp primers: forward: CTT CTTCAAGGACGACGGCAA, reverse: CTTGAACTCGATGCCCTTCAGC; msGAPDH primers: forward: TGTGAGGGAGATGCTCAGTG, reverse: GGCAT TGCTCTCAATGACAA; huBrca2 primers: forward: TGCCTGAAAACCAGATG ACTATC, reverse: AGGCCAGCAAACTTCCGTTTA; huActin primers: forward: AGAGCTACGTGCCCTGAC, reverse: CGTACAGGTCTTTGCCGGATG; msTFEB primers: forward: AGAACCCCACCTCCTACCAC, reverse: GGACTGTTGGGAG CACTGTT; msTMEM55B primers: forward: CGTACGGAGCCGGTAAACAT, reverse: TGATCGGAGACTGACAGACG; msATP6V11 primers: forward: GATT GGAATGGAGCCCTGTA, reverse: TGCTCAATAACCCGTTTTCC; msHEXA primers: forward: GCCATTACCTGCCATTGTCT, reverse: ACCTCCTTCACA TCCTGTGC; msSQSTM1 primers: forward: CCTTGCCCTACAGCTGAGTC, reverse: CTTGTCTTCTGTGCC GTGC; msHIF1alpha primers: forward: TCAAG TCAGCAACGTGGAAG, reverse: TATCGAGGCTGTGTCGACTG; msGBA primers: forward: TGGGTACCTTCAGCCGTTAC, reverse: GAGTAGGTGGGGAC AAAGCA; msHnrnpc primers: forward: TGCAGAGCCAAAAGTGAA, reverse: CACTTTTTGCCCCTTCGTGAA; msIrf7 primers: forward: AAACCATAGAGGC ACCCAAG, reverse: CCCAATAGCCAGTCTCCAAA; msTrp53 primers: forward: CCATCCTGGCTGTAGGGTAGC, reverse: CAGACCAAGAGGCTGAGTCG.

*IRES reporter assay and luciferase-reporter assay.* HCV and CrPV plasmids (Addgene #11510 and #11509, donated by Phil Sharp[65]) were transfected into Atg12^f/f or Atg12^KO MEFs as described above. Ratio of Renilla to Firefly luciferase was assayed using DualGlo Luciferase assay system (Promega E2920) and luminescence (AU) was read by spectrometer.

Luciferase levels for the UTR reporter construct were assayed by Nano-glo luciferase assay system (Promega N1120). GFP levels for the UTR reporter construct were assayed by immunoblot.

*Clonogeneic replating assay.* Cells were grown in control conditions and treated as indicated for 16 h. Following treatment, the cells were trypsinized and 500 live cells were plated in control media, and allowed to grow for 10 days. Colonies were fixed in 4% PFA, stained by crystal violet and counted.

*Crystal violet assay.* Two thousand cells were plated per well in 96-well plates. At time ($t$) = 0 and 24 h, plates were fixed with 4% paraformaldehyde and stained in 0.3% crystal violet in water for 1 h, and washed in distilled water until control empty wells were rinsed clean. The crystal violet stain was solubilized in 100% methanol, and A590 measured by spectrometer. Percent growth (mean ± SEM) was calculated as (Abs.t24 − Abs.t0)/Abs.t0.

*Cell-cycle analysis.* Cells were trypsinized and fixed in ice-cold methanol and stored at −20 °C. Cells were stained in 3.8 mM sodium citrate, 25 μg/ml propidium iodide, and 10 μg/ml RNase A in PBS. Flow was performed on an LSRII SORP machine and analysis of percent of cells in various cell-cycle stages was performed using FlowJo. Flow cytometry data was generated in the UCSF Parnassus Flow Cytometry Core which is supported by the Diabetes Research Center (DRC) grant, NIH P30 DK063720.

*Metabolomics.* Six million cells per condition ($n = 4$) were washed in PBS and pelleted before being snap frozen. Gas chromatography time-of-flight mass spectrometry with the silylation reagent N-tert-butyldimethylsilyl-N-methyltrifluoroacetamide (GC-TOF with MTBSTFA) was performed at the West Coast Metabolomics Center at the University of California, Davis.

*Statistics and reproducibility.* Statistical analysis of ribosome profiling data was performed using Babel[20]. Details of statistical analyses of experiments and number of biological replicates ($n$) can be found in the figure legends. Prior to statistical analysis, outliers were identified by the Dixon test and at most a single value ($p < 0.05$) was excluded from the statistical test, but is still represented in the plots. Wherever an outlier was excluded is stated in the figure legend. Unless otherwise stated, statistical significance was assessed using the two-sample equal variance $t$ test with a cutoff of 0.05 for significance. Normal distribution was assumed unless otherwise stated. Statistical testing was performed in Excel, R, or Prism 8. Boxplots were generated in R using ggplot2.boxplot, producing boxplots in the style of Tukey, where the center line is the median, the lower and upper hinges correspond to the first and third quartile, the upper whisker extends from the hinge to the largest value no further than 1.5* IQR from the hinge (where IQR is the interquartile range, or distance between the first and third quartiles), and the lower whisker extends from the hinge to the smallest value at most 1.5 * IQR of the hinge. Immunoblotting experiments were performed a minimum of three times with independent biological replicates. All raw data are included in Supplementary Data 1.

**Reporting summary.** Further information on research design is available in the Nature Research Reporting Summary linked to this article.

## Data availability
Figure 3 has associated raw data from ribosome profiling and RNAseq. RNAseq and ribosome profiling sequencing data is available at https://www.ncbi.nlm.nih.gov/bioproject/PRJNA634689, Accession: PRJNA634689 ID: 634689. All other Figures have associated raw data available in Supplementary Data 1 (Source data for all plots) and in the Supplementary Information file (Unprocessed immunoblot images). Any additional data, hard copies of lab notebooks stored in the pathology department at UCSF and all data backed up on multiple hard drives and university cloud storage, can be made available upon request by contacting the corresponding author.

## Code availability
All data and code used to analyze the sequencing data are available in UCSF Dash (dayadyrad.org) with the identifier https://doi.org/10.7272/Q6N877ZT.

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

## Acknowledgements

Grant support includes the NIH (R01AG057462, R01CA213775, R01CA126792 to J.D., P30CA082103 to A.O.), QB3/Calico Longevity Fellowship (to J.D.), Samuel Waxman Cancer Research Foundation (to J.D.), and the DOD BCRP (W81XWH-11-1-0130 to J.D.). J.G. was supported by the NSF GRFP (DGE-1144247). T.M. was supported by the NIH (NCI 1F31CA217015. A.M.L. is supported by a Banting Postdoctoral Fellowship from the Government of Canada (201409BPF-335868) and a Cancer Research Society Scholarship for Next Generation of Scientists (22805).

## Author contributions

Conceptualization: J.G. and J.D.; investigation: J.G.; visualization: J.G.; formal analysis: J.G.; data curation: J.G.; writing—original draft: J.G.; writing—review and editing: J.D., T.M., S.A., and A.O.; software: S.A. and A.O.; resources: T.M., D.S., A.M.L., A.O., and S.A.; funding acquisition: J.D. and A.O.; supervision: J.D.

## Competing interests

J.D. serves on the Scientific Advisory Board for Vescor Pharmaceuticals, LLC. All other authors declare no competing interests.
