## [Peer Review File · Communications Biology]

Reviewers' comments:

Reviewer #1 (Remarks to the Author):

In this report Goldsmith, et al study the impact of blocking the autophagy mechanism on the mRNA translation machinery in mammalian cells. Using a genome-wide ribosome profiling technique as well as low-throughput assays, they identify Brca2, as a cellular mRNA, translation of which is upregulated by autophagy mechanism. They further show that BRCA2 expression is required for preventing DNA damage in both cell models as well as an inducible autophagy-deficient mouse model. Overall, this is an interesting study with significant potential to improve our understating of the upstream mechanisms that regulate the mRNA translation machinery and their impact of cell physiology in higher eukaryotes. Nevertheless, there are some important issues with either the design of the study or interpretation of the data that need to be addressed. An important and recurrent issue throughout the manuscript is the lack of clarity in how and why the authors would consider an observation significant or otherwise. I'll summarize a few comments and suggestions on the manuscript below:

1. The phrase "protein translation" is inaccurate. It is better to use either "mRNA translation" or protein synthesis".
2. In figure 1A, there is a clear increase of 35S-incorporation in HBSS-treated cells compared with control. Could the authors explain why despite sever starvation, the cells actually have higher rate of protein synthesis. Also in the same figure, the authors claim they "found no differences in 35S methionine incorporation in Atg12KO cells compared to control (Atg12f/f) cells, in either fed or starved conditions". However, with the resolution of the figure 1A available to me, there seems to be a clear reduction in the intensity of smear in lane 4 compared with lane 3. Similar pattern could be observed for Atg12 and Atg7-depleted cells in Figure S1D. Could the authors explain this discrepancy in the results and conclusion? This point is particularly noticeable when the authors consider the almost negligible difference in BRCA2 expression in Figure 3A and 3E as significant.
3. There is a clear increase in the phosphorylation level of eIF2-alpha in the HBSS cells, which is expected, and it seems to be increased upon Atg12 depletion. Yet the authors claim there is only a slight but not significant increase in eIF2-alpha phosphorylation. But it is not clear where the cut-off for significant is? Did the authors quantify the bands?
4. Unfortunately, I don't seem to have access to the Supplementary Table 1 which shows the list of differentially translated mRNAs. Nor did the authors explain in the manuscript what were the cut-off that they used (p-value, FDR, fold-change, etc) in considering what mRNAs are considered differentially translated. This should be clearly stated in the manuscript. Similar issue exists for the GO analysis in Figure S2D. Is the cut-off p-value < 0.05? Would that be really considerable in such analyses?
5. To study the impact of 5' UTR on translation, the authors used EV vs Brca2 5' UTR, which only proves that addition of Brca2 5' UTR suppresses translation. But is this effect of autophagy depletion on translation, specific to the Brca2 5' UTR? How about Brca1 5' UTR? This assay needs to be repeated with additional target and control mRNAs, in order to provide more conclusive evidence on the impact of autophagy on translation, mediated by 5' UTRs.
6. In Figure 5D, NAC clearly reduces the amount of γ H2AX in WT cells but has no impact on the ATG12-depleted cells. If, according to the authors, the impact of ROS on DNA damage is independent of ATG12, why there is such stark difference between WT and autophagy-deficient cells? Shouldn't the amount of γ H2AX be similar in response to NAC, regardless of ATG12 status?
7. The pattern of γ H2AX expression in the WT vs. ATG12-KO cells is not consistent throughout manuscript. For instance, in lane 1 & 2 of Figure 5E, which are MEFs treated with vehicle, there is no visible difference in γ H2AX expression. Could the authors explain this discrepancy?
8. The quality of WB in Fig. 6E, particularly for Kidney samples are poor and visually not consistent with the quantitation, presented in Fig. 6F. Frankly, in the blot presented in this figure, there is hardly a clear pattern of correlation between Atg12-depletion and BRCA2 expression. In the 4 samples on the left, the reduced BRCA2 expression could just as well be explained by less loading,

according to the Ponceau staining and in the blot on the right, there is hardly any discernible pattern in BRCA2 expression. The difference in Cereberal cortex might be slightly more visible, although the quality of the image for Ponceau staining is very low and makes it difficult to assess the loading levels. I suggest providing better blots and also showing the actual numbers acquired by quantitation for each band for a better evaluation by the readers.

9. In general, the authors appropriately spent a significant portion of the manuscript on deducing the effect of reduced expression of Brca2 on DNA damage in autophagy-depleted cells. However, this seems to have come at the expense of further investigation into how Atg12 expression impacts translation machinery. Frankly, there seems to be no attempt at corroborating the multiple observations that the authors had on cap-binding of different eIF4F components, mTOR activity, reporter assays, or expression of RBPs, in order to come up with a more coherent narrative of how autophagy would impact translation. I suggest the authors further take advantage of the Discussion part of the manuscript and try to collect their thought and give their audience a more nuanced analysis of their findings, rather than just suggesting the presence of additional regulatory mechanisms!

Reviewer #2 (Remarks to the Author):

The manuscript by Goldsmith et al provides new insight into pathways that are translationally regulated by autophagy and is an important contribution that could help explain fundamentally important questions in the field such as how autophagy protects against cancer development. Using ribosome profiling following genetic ablation of the autophagy regulator ATG12, the authors reveal a new function of autophagy in mammalian cells– selective control of translation of a subset of mRNAs. Importantly, this affects critically important cellular functions such as cell cycle control and DNA repair via BRCA2; this may help to explain autophagy’s importance in maintaining genome integrity. The authors perform experiments to test if these effects are mimicked by inactivation of other components of the autophagy conjugation machinery and this therefore seems to be an autophagy-specific effect and show using a nice in vivo model that BRCA2 levels and associated DNA damage increases upon Atg12 inactivation in vivo. These effects are due to alterations in RNA binding proteins in the 5’UTR.

I found the data to be convincing and I have only very minor suggestions/questions:

1. In Fig 6E the Ponceau stains look a bit messy- do the authors have another loading control they could show for these data?
2. The authors show that MSI1 has a LIR and interacts with LC3 orthologues suggesting that it is turned over by selective autophagy. Does it show other attributes of an autophagy substrate- e.g. does it accumulate if you inhibit the lysosome with something like BafA1?
3. The authors state that turnover of BRCA2 is unaltered based on Atg12 status. This is based on data shown in Fig 3H and I. But to me it looks like at the 6 hour time point there is a reduction in the KO cells that is not apparent in the WT controls. Is this real? If so maybe reword the text.

In this report Goldsmith, et al study the impact of blocking the autophagy mechanism on the mRNA translation machinery in mammalian cells. Using a genome-wide ribosome profiling technique as well as low-throughput assays, they identify Brca2, as a cellular mRNA, translation of which is upregulated by autophagy mechanism. They further show that BRCA2 expression is required for preventing DNA damage in both cell models as well as an inducible autophagy-deficient mouse model. Overall, this is an interesting study with significant potential to improve our understating of the upstream mechanisms that regulate the mRNA translation machinery and their impact of cell physiology in higher eukaryotes. Nevertheless, there are some important issues with either the design of the study or interpretation of the data that need to be addressed. An important and recurrent issue throughout the manuscript is the lack of clarity in how and why the authors would consider an observation significant or otherwise. I'll summarise a few comments and suggestions on the manuscript below:

1. The phrase “protein translation” is inaccurate. Either “mRNA translation” or protein synthesis”.
2. In figure 1A, there is a clear increase of ³⁵S-incorporation in HBSS-treated cells compared with control. Could the authors explain why despite sever starvation, the cells actually have higher rate of protein synthesis. Also in the same figure, the authors claim they “found no differences in ³⁵S methionine incorporation in Atg12^{KO} cells compared to control (Atg12^{f/f}) cells, in either fed or starved conditions”. However, with the resolution of the figure 1A available to me, there seems to be a clear reduction in the intensity of smear in lane 4 compared with lane 3. Similar pattern could be observed for Atg12 and Atg7-depleted cells in Figure S1D. Could the authors explain this discrepancy in the results and conclusion? This point is particularly noticeable when the authors consider the almost negligible difference in BRCA2 expression in Figure 3A and 3E as significant.
3. There is a clear increase in the phosphorylation level of eIF2-alpha in the HBSS cells, which is expected, and it seems to be increased upon Atg12 depletion. Yet the authors claim there is only a slight but not significant increase in eIF2-alpha phosphorylation. But it is not clear where the cut-off for significant is? Did the authors quantify the bands?
4. Unfortunately, I don't seem to have access to the Supplementary Table 1 which shows the list of differentially translated mRNAs. Nor did the authors explain in the manuscript what were the cut-off that they used (p-value, FDR, fold-change, etc) in considering what mRNAs are considered differentially translated. This should be clearly stated in the manuscript. Similar issue exists for the GO analysis in Figure S2D. Is the cut-off p-value<0.05? Would

that be really considerable in such analyses?

5. To study the impact of 5'UTR on translation, the authors used EV vs Brca2 5' UTR, which only proves that addition of Brca2 5'UTR suppresses translation. But is this effect of autophagy depletion on translation, specific to the Brca2 5' UTR? How about Brca1 5'UTR? This assay needs to be repeated with additional target and control mRNAs, in order to provide more conclusive evidence on the impact of autophagy on translation, mediated by 5' UTRs.
6. In Figure 5D, NAC clearly reduces the amount of γ H2AX in WT cells but has no impact on the ATG12-depleted cells. If, according to the authors, the impact of ROS on DNA damage is independent of ATG12, why there is such stark difference between WT and autophagy-deficient cells? Shouldn't the amount of γ H2AX be similar in response to NAC, regardless of ATG12 status?
7. The pattern of γ H2AX expression in the WT vs. ATG12-KO cells is not consistent throughout manuscript. For instance, in lane 1 & 2 of Figure 5E, which are MEFs treated with vehicle, there is no visible difference in γ H2AX expression. Could the authors explain this discrepancy?
8. The quality of WB in Fig. 6E, particularly for Kidney samples are poor and visually not consistent with the quantitation, presented in Fig. 6F. Frankly, in the blot presented in this figure, there is hardly a clear pattern of correlation between Atg12-depletion and BRCA2 expression. In the 4 samples on the left, the reduced BRCA2 expression could just as well be explained by less loading, according to the Ponceau staining and in the blot on the right, there is hardly any discernible pattern in BRCA2 expression. The difference in Cerebral cortex might be slightly more visible, although the quality of the image for Ponceau staining is very low and makes it difficult to assess the loading levels. I suggest providing better blots and also showing the actual numbers acquired by quantitation for each band for a better evaluation by the readers.
9. In general, the authors appropriately spent a significant portion of the manuscript on deducing the effect of reduced expression of Brca2 on DNA damage in autophagy-depleted cells. However, this seems to have come at the expense of further investigation into how Atg12 expression impacts translation machinery. Frankly, there seems to be no attempt at corroborating the multiple observations that the authors had on cap-binding of different eIF4F components, mTOR activity, reporter assays, or expression of RBPs, in order to come up with a more coherent narrative of how autophagy would impact translation. I suggest the authors further take advantage of the Discussion part of the manuscript and try to collect

their thought and give their audience a more nuanced analysis of their findings, rather than just suggesting the presence of additional regulatory mechanisms!

Goldsmith et al. found that global protein synthesis and amino acid pool are largely intact in the absence of autophagy in mammalian cells unlike *Saccharomyces cerevisiae*. However, using ribosome profiling, they found that autophagy controls some specific group of mRNAs that are related to cell cycle control and DNA damage repair, including BRCA2. Mechanistically, autophagy controls translation of BRCA2 through its complex 5' UTR and RNA-binding protein including eIF4A1 and MSI1. Like BRCA2 deficient cells, autophagy deficient cells and mice exhibited increased levels of γ H2AX and hypersensitivity to PARP inhibitors, as well as formation of abnormal centrosomes. They found a new role for autophagy in the control of translation of specific target RNAs. Their findings are novel and the manuscript is well written, but I would like to ask the authors to address the following concerns before publication.

Major points:

1. The authors showed that autophagy deficient cells produce a reduced level of BRCA2 and that the autophagy deficient cells and mice exhibit the phenotypes that are reminiscent of BRCA2 deficient ones. However, it is not clear whether the autophagy phenotypes are caused by a reduced amount of BRCA2 or not. It seems that overexpression of human BRCA2 canceled the effect of ATG12 depletion on γ H2AX expression (Fig. 5D graph). However, the BRCA2 overexpression itself increased γ H2AX levels even in the presence of intact autophagy compared to the control experiments, making it difficult to conclude whether the BRCA2 overexpression rescued the autophagy phenotype or not. Can the authors examine the effect of the ectopic expression of BRCA2 on other phenotypes of autophagy deficient cells and/or mice. The section title in the results “Decreased BRCA2 results in DNA damage accumulation and centrosome defects in autophagy deficient cells” is too strong at the present stage.
2. The authors argue that “MSI1 knockdown partially recovered BRCA2 levels in *Atg12^{KO}* MEFS (Figure 4J, K)”. Because these are important data to understand the

control of BRCA2 translation, can the authors repeat the experiments shown in Fig. 4J and K, measure the band intensity, and carry out statistical analysis.

Minor points:

1. Can the authors provide the statistical explanation of “no difference” (page 6) and “no changes” (page 7).
2. Can the authors clarify whether multiple rounds of immunostaining have been done using the same blots or different blots in the same panel (e.g. Fig. 1A).
3. Some characters are too tiny to see (e.g. Fig. 1A, 64kDa; Fig. 1H, K; Fig. 2A-F; Fig. 3A,C,E; Fig. 4A,E,H,J,K; Fig. 5D,E.; Fig. 6B,E).
4. Some differences are not very clear without statistical evaluation. Can the authors provide P-values in Fig. 3B,D,F,K, Fig. 4G,H, Fig. 5D,F, Fig. S2E, Fig. S4A,E,G,H,I, and Fig. S5B,D,E.
5. Can the authors count the nuclei containing γ H2AX and/or 53BP1, and perform statistical analysis to show the effect of ATG12 KO in Fig. 5A-B.
6. For some abbreviations, it is better to spell out when they appear for the first time in the text (e.g. IRES and TOP).
7. Fig. 1H “p-4EBP” and “1 S65” must be “p-4EBP1” and “S65”.
8. In Fig. 6B, Atg12 + or – labeling and/or the p62 panels seem incorrect.
9. Would you please provide Supplementary Table 1.

POINT-BY-POINT RESPONSE FOR COMMSBIO-19-0971-T

We thank the reviewers for their comments. To summarize, the reviewers brought up 3 major concerns:

1. further clarify specific mechanisms of how autophagy affects translation machinery
2. better describe statistical tests
3. provide clearer images and quantifications of certain western blots

To facilitate reviews, we have embedded data from the new figures in the revised manuscript, as well as included additional data addressing specific concerns raised by the reviewers in the response below.

Reviewer #1:

We thank Reviewer 1 for taking the time to critically read our paper, and we appreciate his/her expert input regarding translation, particularly as we have extended into the field of RNA translation from autophagy to write this multidisciplinary paper. We are incredibly pleased that the reviewer found our study interesting, with significant potential to improve our understanding of upstream mechanisms regulating mRNA translation machinery. In order to improve the paper, we endeavored to address and correct every point Reviewer 1 brought up.

1. the phrase “protein translation” is inaccurate.

We apologize for use of this inaccurate term. We have corrected all instances of this throughout the text, replacing it with the phrase “mRNA translation” or “protein synthesis”.

2A. In Figure 1A, there is a clear increase of ³⁵S-incorporation in HBSS-treated cells compared with control. Could the authors explain why despite severe starvation, the cells actually have higher rates of protein synthesis?

Indeed, this is a counterintuitive, but a highly consistent and robust finding that we have been working to further understand. We have observed this increase in protein synthesis upon severe starvation induced by HBSS treatment, a near universal method of starvation used by the autophagy field, in a large number of cell lines, both primary and immortalized. We are preparing a short manuscript describing this finding and further characterizing the requirements. In order to be completely transparent about this result, we have made a serious effort to describe in great detail the exact methods used for this assay, so that others will be able to replicate this result.

We now also include new data in Figure 1 (Figure 1C, D) as well as brief discussion of the results in the manuscript text, to help clarify this point. We have assayed protein synthesis rates by puromycin incorporation measured with an anti-puromycin antibody. While puromycin incorporation has the major caveat that it terminates translation as it incorporates, we found that the rate of puromycin incorporation during HBSS starvation decreased, as expected, but there was again no difference in the rate of incorporation between Atg12^{ff} and Atg12^{KO} MEFs, statistically tested by ANOVA with Tukey’s HSD post hoc test. It is also to remember that the main goal of these experiments is to compare the rates of translation in control versus autophagy deficient cells. Accordingly, using either ³⁵S methionine incorporation or puromycin incorporation, we have never observed any differences in Atg12^{KO} cells compared to control (Atg12^{ff}) cells, in either fed or starved conditions.

2B. Also in the same figure, the authors claim they “found no differences in ³⁵S methionine incorporation in Atg12KO cells compared to control (Atg12f/f) cells, in either fed or starved conditions”. However with the resolution of the figure 1A available to me, there seems to be a clear reduction in the intensity of the smear in lane 4 compared with lane 3. Similar pattern could be observed for Atg12 and Atg7-depleted cells in Figure S1D. Could the authors explain this discrepancy in the results and conclusion? This point is particularly noticeable when the authors consider the almost negligible difference in BRCA2 expression in Figure 3A and 3E as significant.

We thank the reviewer for bringing this important point for clarification to our attention. As the reviewer notes, because the difference we see in total BRCA2 protein levels is small, it is important to ensure that the change in the rate of protein synthesis of BRCA2 is not in fact due to small changes in the overall rate of protein synthesis. We quantified 11 biological replicates for this experiment, as well as 5 replicates in primary MEFs, and show the data plotted in Figure 1B and S1E. We found that despite some variation between experiments, there was no statistically significant difference in the rate of ³⁵S methionine incorporation by ANOVA with Tukey’s HSD post hoc test, which we now clearly include in the graph and figure legend.

Furthermore, although the changes in the total amount of BRCA2 present in Atg12KO cells are very small and difficult to assay (due to the technical challenge of BRCA2 being a very large protein with poor antibody reagents), we feel confident that 1) the rate of BRCA2 synthesis is indeed lower in Atg12^{KO} cells based on the AHA incorporation data in Figure 3 (now 3A and 3B) and 2) this resultant small decrease in the total levels of BRCA2 has a measurable physiological impact, shown in the results from Figure 6 (previously Figure 5).

3. There is a clear increase in the phosphorylation level of eIF2-alpha in the HBSS cells, which is expected, and it seems to be increased upon Atg12 depletion. Yet the authors claim there is only a slight but not significant increase in eIF2-alpha phosphorylation. But it is not clear where the cut-off for significance is? Did the authors quantify the bands?

Yes, we quantified the bands for eIF2-alpha phosphorylation from multiple immunoblot experiments, which are presented in Figure 11. We performed an ANOVA with Tukey’s HSD post-hoc test, and found that the p-value between Atg12f/f and Atg12KO cells in the HBSS condition was greater than 0.05, and thus we previously stated that the change was not significant. In the revised manuscript we have now fully clarified in the text how significance has been analyzed, and provide the exact p-values in the figure legend for Figure 11; our analysis

indicate that p-values for Atg12f/f samples compared to Atg12KO samples in control media or HBSS treatment are $p = 0.99$ and $p = 0.85$ respectively.

4. Unfortunately, I don't seem to have access to the Supplementary Table 1 which shows the list of differentially translated mRNAs. Nor did the authors explain in the manuscript what were the cut-off that they used (p-value, FDR, fold-change, etc) in considering what mRNAs are considered differentially translated. This should be clearly stated in the manuscript. Similar issue exists for the GO analysis in Figure S2D. Is the cut-off $p\text{-value} < 0.05$? Would that be really considerable in such analyses?

We apologize for not uploading Supplementary Table 1 (now Supplementary Table 2), which we have included now. Furthermore, although we stated the p-value threshold levels for analysis of the ribosome profiling data in the figure legend, we apologize for not clearly stating it in the text as this is a crucial point. We used a p-value threshold level of $p < 0.005$ for the reduced occupancy group and $p < 0.01$ for the increased occupancy group, which we now state in the main body text.

For the GO analysis in Figure S2D, we plotted a p-value cut-off of < 0.05 as shown. This may not be of strict statistical significance, and so we have amended the text to reflect this. However, in the revised manuscript, we have confirmed Atg12 deletion affects cell cycling rates, which correlates with the GO analysis in Figure S2D. While we remain circumspect whether these changes in the translation of cell cycle mRNAs are the sole cause of slowed progression through the cell cycle in ATG12KO cells, we have now included this data in Figure S2E-H.

5. To study the impact of 5'UTR on translation, the authors used EV vs Brca2 5' UTR, which only proves that addition of Brca2 5' UTR suppresses translation. But is this effect of autophagy depletion on translation, specific to the Brca2 5' UTR? How about Brca1 5' UTR? This assay needs to be repeated with additional target and control mRNAs, in order to provide more conclusive evidence on the impact of autophagy on translation, mediated by 5' UTRs.

This is an excellent point. Upon further analysis during the revision period, we do not believe that the impact of autophagy on translation is solely mediated by 5'UTRs. We now include new results (Figure 5 and S5) describing how certain RNA-binding proteins downstream of autophagy may impact the translation of other mRNAs based on the motif search in the sequences of the UTRs from mRNAs identified by ribosome profiling. Indeed, we find a large number of RBP motif sequences located at both 5' and 3' UTRs of the significant hits from our RP. Interestingly, many of these RBPs have been proposed to bind to LC3/ATG8 orthologues, broaching their regulation via the autophagy pathway. However, further scientific validation of each of these RBPs and their mechanism of action via 5' vs 3' UTRs is outside of the scope of this paper. This type of study identifying critical motifs that make different UTRs, both 5' and 3', sensitive to autophagy inhibition could be a paper in its own right, and it would not be feasible to perform within the reasonable timeline of revisions requested for this paper. Based on this new data, we endeavor to not overstate our findings in the main text of the paper regarding 5'UTRs, and limit our statements to the effect of autophagy on the translation of *Brca2* itself. In the revised discussion, we have tempered our conclusions regarding the 5'UTR. We state: *"In addition to the autophagy-dependent effects on the 5'UTR of Brca2, we recognize that a broader repertoire of translation control mechanisms are impacted by autophagy, including the modulation of additional LC3/ATG8-binding RBPs, such as eIF4A1, MATR3 and YBX1... Overall, identifying the diverse array of molecular mechanisms by which autophagy impacts mRNA translation, both directly and indirectly, remains an important area for further study."*

Figure 5

Figure S5

6. In Figure 5D, NAC clearly reduces the amount of γ H2AX in WT cells but has no impact on the ATG12-depleted cells. If, according to the authors, the impact of ROS on DNA damage is independent of ATG12, why there is such stark difference between WT and autophagy-deficient cells? Shouldn't the amount of γ H2AX be similar in response to NAC, regardless of ATG12 status?

NAC is highly acidic, and significantly alters the pH of the media upon addition. Although NAC is a commonly used ROS scavenger, in our experiments, we cannot be certain that this change in pH to cells has unintended consequences. In the original manuscript, we provided the quantification of the four biological experiments that we have plotted above the immunoblot. While in one experiment, we do see decreased levels of γ H2AX in the NAC treated 12^{ff} cells, overall we do not find a consistent decrease across multiple bio-replicates.

We believe that the more biologically relevant comparison is the change in γ H2AX levels between $Atg12^{ff}$ and $Atg12^{KO}$ between groups. In such cases, we consistently see that the $Atg12^{KO}$ cells (control, pGFP and +NAC) have roughly twice as much γ H2AX, while the BRCA2 overexpressing $Atg12^{KO}$ cells have the same levels of γ H2AX as the $Atg12^{ff}$ cells. We have clarified this point in the text, and now include p-values in the figure (now figure 6D) to help evaluate the changes.

D.

7. The pattern of γ H2AX expression in the WT vs. ATG12-KO cells is not consistent throughout manuscript. For instance, in lane 1 & 2 of Figure 5E, which are MEFs treated with vehicle, there is no visible difference in γ H2AX expression. Could the authors explain this discrepancy?

Once again, it is important not to place excessive emphasis on the single immunoblotting image in Figure 6E (previously Figure 5E) without more broadly considering the quantifications presented in Figure S6F (previously Figure S5E). In each experiment, there is some variability in how much γ H2AX is detected. We have attempted to minimize technical variability as much as possible by consistent methods that we have clearly described in the methods. Because immunoblotting is variable, the quantification averaged over multiple biological replicates is critical to evaluate. In this particular immunoblot in question, because the γ H2AX signal in the PARP inhibitor treated samples is much greater, it overwhelms the signal in the untreated control samples. In order to clearly show the γ H2AX signal in the PARP inhibitor treated samples, the exposures do not allow us to fully delineate the γ H2AX signals in the $Atg12^{ff}$ versus $Atg12^{KO}$ control (non-PARP inhibitor treated) samples. Furthermore, we show that not

only is the γ H2AX signal elevated, this has a functional consequence as the cells are impaired in colony forming ability, shown in Figure 6F (previously Figure 5F).

Nevertheless, to allay the reviewer's concerns, we re-ran samples to provide a cleaner blot, which we have replaced in the revised manuscript.

E.

8. The quality of WB in Fig. 6E, particularly for Kidney samples are poor and visually not consistent with the quantitation, presented in Fig. 6F. Frankly, in the blot presented in this figure, there is hardly a clear pattern of correlation between Atg12-depletion and BRCA2 expression. In the 4 samples on the left, the reduced BRCA2 expression could just as well be explained by less loading, according to the Ponceau staining and in the blot on the right, there is hardly any discernible pattern in BRCA2 expression. The difference in Cerebral cortex might be slightly more visible, although the quality of the image for Ponceau staining is very low and makes it difficult to assess the loading levels. I suggest providing better blots and also showing the actual numbers acquired by quantitation for each band for a better evaluation by the readers.

We have re-run the samples in order to provide a clearer image of the loading controls. We apologize for the quality of the immunoblot images; BRCA2 is a notoriously difficult protein to study because of its large size and a dearth of commercially available antibodies. However, we have attempted to improve upon the images presented. We feel however that this modest change BRCA2 levels observed in tissues in Atg deleted mice is 1) consistent with the small decrease in total BRCA2 observed in cell lines and 2) important because it explains many of the phenotypes previously linked to autophagy deletion, including higher levels of DNA damage.

E.

9. In general, the authors appropriately spent a significant portion of the manuscript on deducing the effect of reduced expression of Brca2 on DNA damage in autophagy-depleted cells. However, this seems to have come at the expense of further investigation into how Atg12 expression impacts translation machinery. Frankly, there seems to be no attempt at corroborating the multiple observations that the authors had on cap-binding of different eIF4F components, mTOR activity, reporter assays, or expression of RBPs, in order to come up with a more coherent narrative of how autophagy would impact translation. I suggest the authors further take advantage of the Discussion part of the manuscript and try to collect their thought

and give their audience a more nuanced analysis of their findings, rather than just suggesting the presence of additional regulatory mechanisms!

To address this major criticism offered by reviewer 1, we have performed several experiments and revised our discussion section to address this concern. We expanded our paper to include new figures (Figure 4G-N, Figure S5).

Figure 4

Briefly, we make the case that the accumulation of p62/SQSTM1 in autophagy deficient cells leads to the sequestration of eIF4A1, in a model similar to that described of p62/SQSTM1 sequestering KEAP. We believe that the decrease in available eIF4A1, contributes to the changes in translation of certain mRNAs.

Furthermore, we describe how autophagy deficiency affects other the levels or functions of other RNA binding proteins in the cell. The combined analysis of the autophagy associated proteome (Behrends et al, 2010) and RNA binding proteins (Castello et al, 2012) shows a large percent of RNA binding proteins are associated with autophagy adaptors. Using RBPDB (<http://rbpdb.cabr.utoronto.ca>) we performed an RNA binding protein motif search in the UTRs of the mRNAs that appear to have altered translation when autophagy is inhibited. We have found MATR3 and YBX1 binding sites to be associated with alterations in ribosome occupancy upon autophagy deletion, and these RBPs associate with some Atg8/LC3 family autophagy adaptors. This suggests that impaired translation due to increased MS11 following autophagy inhibition is specific to of *Brca2*, and the combination of changes of other RNA binding proteins,

including eIF4A1, MATR3 and YBX1, may regulate other mRNAs identified by ribosome profiling.

Figure S5

By addressing these concerns with further experiments, we feel that we have both confirmed and strengthened our results, and we thank the reviewer for suggesting that we include this data.

Reviewer #2:

We thank Reviewer 2 for his/her helpful comments, and we appreciate that Reviewer 2 found our manuscript to be of importance.

1. In Fig 6E the Ponceau stains look a bit messy- do the authors have another loading control they could show for these data?

We apologize for the poor loading. Because of the size of BRCA2, we were unable to blot for the usual loading controls on the same gel as the BRCA2 quantifications. We have re-run the lysates to include better images of loading controls (now Fig7E).

2. The authors show that MSI1 has a LIR and interacts with LC3 orthologues suggesting that it is turned over by selective autophagy. Does it show other attributes of an autophagy substrate- e.g. does it accumulate if you inhibit the lysosome with something like BafA1?

While we did not show accumulation with Bafilomycin A1 in the original manuscript, we do show that MSI1 accumulates upon Atg12 deletion. However, it does not appear to decrease upon autophagy induction by starvation (Figure 5B). As the reviewer requests, have performed new experiments treating control MEFs with the lysosomal inhibitors BafA1 and Chloroquine for 4 hours, but do not observe accumulation of MS1 in response to these treatments.

There are several possible reasons for this result. First, the actual pool of MS1 within a cell subject to selective autophagic turnover is minor and the methods we have employed analyzing steady state proteins levels are not sensitive enough to detect low level turnover of this protein. Second, MS1 may be selectively captured and incorporated into the early autophagosome, thereby sequestering it away from mRNA, but the protein itself is not efficiently degraded over the duration that we have analyzed. Indeed, we have previously reported that LC3 binding and capture into autophagosomes can titrate away the protein TBC1D5, a negative regulator of the retromer complex, this RABGAP proteins is not efficiently degraded via autophagy. Nonetheless, autophagy-dependent shuttling of TBC1D5 into LC3 compartments is crucial for retromer-dependent cell surface trafficking of GLUT1 and other proteins (Roy, Molecular Cell 2017, 67(1):84-95, PMID: 28602638). Although interesting, further analysis of these possibilities is well beyond the primary focus of this manuscript.

3. The authors state that turnover of BRCA2 is unaltered based on Atg12 status. This is based on data shown in Fig 3H and I. But to me it looks like at the 6 hour time point there is a reduction in the KO cells that is not apparent in the WT controls. Is this real? If so maybe reword the text.

We performed statistical analysis on the quantification of the immunoblots and found no statistically significant change ($p > 0.05$) for all of the time points analyzed. We have amended the text to clarify this point.

Reviewer #3:

We thank Reviewer 3 for taking the time to critically address our manuscript, and for his/her thoughtful comments. We endeavor to address all of them below.

Major points:

1. The authors showed that autophagy deficient cells produce a reduced level of BRCA2 and that the autophagy deficient cell sand mice exhibit phenotypes that are reminiscent of BRCA2 deficient ones. However, it is not clear whether the autophagy phenotypes are caused by a reduced amount of BRCA2 or not. It seem that overexpression of human BRCA2 cancelled the effect of ATG12 depletion on γ H2AX expression. However the BRCA2 overexpression itself increased γ H2AX levels, even in the presence of intact autophagy compared to the control experiments, making it difficult tot conclude whether the BRCA2 overexpression rescued the autophagy phenotype or not.

Indeed, we have found that overexpression of any protein in MEFs is sufficient to elicit an increase in γ H2AX levels. As a result, to control for the effects of overexpression, we have analyzed cells ectopically expressing GFP as a negative control. When we compare $Atg12^{ff}$, the levels of γ H2AX are similar in the GFP overexpressing cells to the BRCA2 overexpressing cells, however, in the $Atg12^{KO}$ cells, the BRCA2 overexpressing cells have lower levels of γ H2AX compared to the GFP overexpressing cells.

Can the authors examine the effect of the ectopic expression of BRCA2 on other phenotypes of autophagy deficient cells and/or mice. The section title in the results “Decreased BRCA2 results in DNA damage accumulation and centrosome defects in autophagy deficient cells” is too strong at the present state.

We performed immunoblots for LC3 and p62 with the BRCA2 overexpressing cells, and include this data in the supplemental figure (FigS6D). We have changed the section title to “Decreased BRCA2 contributes to DNA damage accumulation and centrosome defects in autophagy deficient cells”.

2.The authors argue that “MSI1 knockdown partially recovered BRCA2 levels in the $Atg12^{KO}$ MEFs.” Because these are important data to understand the control of BRCA2 translation, can the authors repeat the experiments in Figure 4J and K , measure the band intensity and carry out statistical analysis?

We have now included quantification and statistical analysis of the data, now presented in Figure 5D-G and clarified the effects of MS1 depletion in the text. Upon depleting MS1 in $Atg12^{ff}$ and $Atg12^{KO}$ MEFs by shRNA, we observed reductions in the steady state levels of

BRCA2 compared to non-targeting control shRNA in both cell types (Figure 5D, $p=0.23$). Nonetheless, upon MS1 depletion, the reduction in BRCA2 protein levels in Atg12^{KO} compared to Atg12^{fl/fl} MEFs was significantly less pronounced.

We also provide new data in the revision demonstrating that MS1 is only one of several RBPs implicated in the control mRNA translation in autophagy-competent versus deficient cells; others include eIF4A1, YBX1 and MATR3. Based on our results, we postulate that Brca2 translation is partly controlled by the autophagic turnover of MS1, and that the regulation of other mRNAs in an autophagy sensitive manner likely arises from the coordinate regulation of multiple LC3/ATG8-interacting RBPs.

Minor points:

1. Can the authors provide statistical explanation of “no difference” p6 and “no changes” p7.

We thank the reviewer for emphasizing the importance of reporting statistical tests and careful language regarding statistics. For statements of no difference or no changes, we based those off of a p-value greater than 0.05, however this was not previously stated clearly in the text. We provide an explanation of statistical test and report the p-values for the results in question in the figures.

2. Can the authors clarify whether multiple rounds of immunostaining have been done using the same blots or different blots in the same panel (e.g. Figure 1A).

We have now clarified in the figure legend text where the same blot has been probed for multiple antibodies. Regardless, whenever a blot is shown grouped together, it is from the same biological replicate.

3. Some characters are too tiny to see.

We have changed the kDa markers and other characters in the figures to a larger size.

4. Some differences are not very clear without statistical evaluation. Can the authors provide p-values in Figure 3B, D, F, K, Figure 4G, H, Figure 5D, F, Figure S2E, Figure S4A, E, G, H, I and Figure S5B, D, E.

We have now included the p-values plus a description of the statistical test for these figures in the figure legends.

5. Can the authors count the nuclei containing γ H2AX and/or 53BP1, and perform statistical analysis to show the effect of Atg12KO in Figure 5A-B.

We provide quantification of percent of γ H2AX positive cells in 12f/f vs 12KO cells, which is now included in Figure 6A.

6. For some abbreviations, it is better to spell out when they appear for the first time in the text (eg IRES and TOP)

We have amended the text accordingly.

7. Figure 1H “p-4EBP” and “1S65” must be “p-4EBP1” and “S65”

We have amended the figure to correctly report the levels of p-4EBP1 S65 in this panel.

8. In Figure 6B, Atg12 + or – labeling and or the p62 panels seem incorrect.

We thank the reviewer for catching this mistake. Indeed, the labeling above the tissues from 12f/f and 12KO animals was reversed and we have corrected the labelling.

9. Would you please provide Supplementary Table 1.

We apologize for failing to include this Table in the original version. It is included now as Supplementary Table 2.

Reviewers' comments:

Reviewer #1 (Remarks to the Author):

I believe the authors successfully addressed most of the points raised in my initial review through additional experimentation/analyses or revising the text. I therefore think the manuscript should be suitable for publication.

Minor point:

The order of panels I and J in Figure 1 seems to be flipped.

Reviewer #2 (Remarks to the Author):

The authors have addressed my previous concerns.

Reviewer #3 (Remarks to the Author):

I found that Goldsmith et al. responded to all the concerns that I had. However, it turned out that their data (Fig. 3B, 3D, 3F, 3H, 3K, 4H, 4L, 4M, 4N, 5A, 5B, 7F, S1C, S1G, and S4E) do not fully support one of the most important conclusions of this paper that is written in the abstract "In particular, we demonstrate that autophagy enables the translation of the DNA damage repair protein BRCA2. (line 38-39)" I believe that we can argue that the difference or change between two sets of data has a meaning only when it is statistically evaluated ($p < 0.05$). But, the authors sometimes argue the difference even though it is statistically non-significant ($p > 0.05$). I would suggest the authors to reconsider their data more carefully and rewrite the paper. Below are the details.

1. Line 97: "major transcriptional changes associated with starvation¹¹ (Figure S1C)." But, there are no statistical comparison between control and HBSS.
2. Line 113-115: "Only two amino acids, glutamine and glycine, were decreased in Atg12KO cells compared to controls grown in nutrient-rich full media conditions (Figure S1G, H)". But, Fig. S1G shows that there is no statistical difference (no asterisks) between Atg12f/f and Atg12KO.
3. Line 199-201: "We observed impaired label incorporation in the Atg12KO cells compared to control cells (Figure 3A-B), demonstrating a reduced rate of BRCA2 synthesis in Atg12KO cells". However, Fig 3B shows that the p value between scr and Atg12KO is 0.08.
4. Line 203-205: "The decrease in the rate of BRCA2 production in Atg12KO cells correlated with reduced BRCA2 protein levels compared to controls in both fed and starved conditions (Figure 3C, D)." However, Fig 3D shows "n.s." between + and - Atg12 in the starved condition (HBSS).
5. Line 206-207: "we observed lower steady state BRCA2 protein levels in Atg5 deleted and Atg7 depleted MEFs (Figure 3E, F)." However, Fig. 3F shows "n.s." between + and - Atg5/7.
6. Line 207-209: "In addition, CRISPR engineered HEK293T cells lacking Atg7, Atg14, and Atg12 exhibited lower steady state BRCA2 protein levels (Figure 3G, H)". However, Fig. 3H shows "n.s." between scr and Atg12KO-Atg7KO-Atg14KO.
7. Line 213-215: "We found no significant difference between Atg12f/f and Atg12KO MEFs in either Brca2 mRNA levels (Figure 3I) or in BRCA2 protein stability or turnover following cycloheximide treatment (Figure 3J, K)". However, Fig. 3K shows that there seems a significant reduction of BRCA2 in Atg12KO ($p=0.02$) but not in the Atg12f/f control ($p=0.11$).
8. Line 242-244: "Indeed, Irf7, another hit from our ribosome profiling screen showed lower protein levels in Atg12KO cells, is notable for a complex 5'UTR secondary structure 25,26 (Figure S4E)." However, Fig. S4E shows that p values are 0.52 and 0.65 between + and - Atg12, in the control and HBSS cases, respectively.
9. Line 261-263: "We observed increased co-location of eIF4A1 within puncta of the ACR p62/SQSTM1 in autophagy deficient cells (Figure 4G, H)". However, Fig. 4H shows $p = 0.06$.

10. Line 270-271: "However, p62/SQSTM1 knockdown was not sufficient to restore BRCA2 levels in Atg12KO cells (Figure 4L-N)". However, there is lack of statistics in Fig. 4L-N.
11. Line 284-286: "We therefore assayed MSI1 binding to Brca2 by RNA immunoprecipitation and observed increased MSI1 associated with Brca2 in the Atg12KO cells (Figure 5A)." However, Fig. 5A shows $p = 0.17$ between Atg12f/f and Atg12KO.
12. Line 287: "a modest accumulation of MSI1". However, Fig. 5B shows $p = 0.60$ and 0.21 between + and - Atg12, in cases of the control and HBSS, respectively.
13. Line 363-365: "Immunoblotting revealed decreased BRCA2 protein levels in the kidney and cerebral cortex of Atg12KO mice compared to autophagy competent Atg12f/f controls (Figure 7E, F)." However, Fig. 7F shows that there is a significant difference between Atg12f/f and Atg12KO only in Cerebral cortex but not in Kidney.
14. Fig. S2H (new data): It is essential to show the results at $t = 0$ to make sure the synchronization was effective equally in Atg12f/f and Atg12KO.
15. Fig. 4I (new data): The essential control of eIF4A1 immunoprecipitation, the eIF4A1 staining, is missing.
16. I could not find the explanation of Fig. 2A-C and Fig. 5E-F in the Result section.
17. "no difference" should be "no significant difference" (lines 103, 141, and 144).

POINT-BY-POINT RESPONSE FOR COMMSBIO-19-0971

Reviewer #1 (Remarks to the Author):

I believe the authors successfully addressed most of the points raised in my initial review through additional experimentation/analyses or revising the text. I therefore think the manuscript should be suitable for publication.

Thank you for the positive comments and the support for publication of this manuscript.

Minor point:

The order of panels I and J in Figure 1 seems to be flipped.

We have rearranged the figures in order to group them more clearly. We will defer to the editor's decision if they believe the panels should be arranged differently.

Reviewer #2 (Remarks to the Author):

The authors have addressed my previous concerns.

Thank you for the support for publication of this manuscript.

Reviewer #3 (Remarks to the Author):

I found that Goldsmith et al. responded to all the concerns that I had. However, it turned out that their data (Fig. 3B, 3D, 3F, 3H, 3K, 4H, 4L, 4M, 4N, 5A, 5B, 7F, S1C, S1G, and S4E) do not fully support one of the most important conclusions of this paper that is written in the abstract “In particular, we demonstrate that autophagy enables the translation of the DNA damage repair protein BRCA2. (line 38-39)” I believe that we can argue that the difference or change between two sets of data has a meaning only when it is statistically evaluated ($p < 0.05$). But, the authors sometimes argue the difference even though it is statistically non-significant ($p > 0.05$). I would suggest the authors to reconsider their data more carefully and rewrite the paper. Below are the details.

1. Line 97: “major transcriptional changes associated with starvation¹¹ (Figure S1C).” But, there are no statistical comparison between control and BBSS.

We have now included P-values for figure S1C, and a detailed description was included in the figure legend. The interpretation of the data has not changed.

2. Line 113-115: “Only two amino acids, glutamine and glycine, were decreased in Atg12KO cells compared to controls grown in nutrient-rich full media conditions (Figure S1G, H)”. But, Fig. S1G shows that there is no statistical difference (no asterisks) between Atg12f/f and Atg12KO.

We have removed the statement that glutamine levels are lower in ATG12KO cells, in accordance with our results that the statistical significance does not reach below the $p = 0.05$ threshold.

3. Line 199-201: “We observed impaired label incorporation in the Atg12KO cells compared to control cells (Figure 3A-B), demonstrating a reduced rate of BRCA2 synthesis in Atg12KO cells”. However, Fig 3B shows that the p value between scr and Atg12KO is 0.08.

We have included an additional biological replicate to the analysis. Although there is some variability in the data, the results from 4 biological replicates a significant p-value ($p = 0.05$) by t test. The results from the 4 individual experiments are now plotted in a scatter plot.

4. Line 203-205: “The decrease in the rate of BRCA2 production in Atg12KO cells correlated with reduced BRCA2 protein levels compared to controls in both fed and starved conditions (Figure 3C, D).” However, Fig 3D shows “n.s.” between + and – Atg12 in the starved condition (HBSS).

We have performed further statistical analysis, and one point was concluded to be an outlier by the Dixon test ($p = 0.001$). When this data point was removed, the p value between Atg12 WT and KO in HBSS conditions is 0.05. We continue to report all of the data points in the graph, but have specified in the figure legend that the p value for HBSS treated cells is calculated at 0.05 upon removal of outliers identified by the Dixon test.

5. Line 206-207: “we observed lower steady state BRCA2 protein levels in Atg5 deleted and Atg7 depleted MEFs (Figure 3E, F).” However, Fig. 3F shows “n.s.” between + and – Atg5/7.

6. Line 207-209: “In addition, CRISPR engineered HEK293T cells lacking Atg7, Atg14, and Atg12 exhibited lower steady state BRCA2 protein levels (Figure 3G, H)”. However, Fig. 3H shows “n.s.” between scr and Atg12KO-Atg7KO-Atg14KO.

With regard to comments 5 and 6, we agree with the reviewer that the results in Figures 3E-H of the previous version were not strongly convincing. In particular, the data from shAtg7 MEFs was not supportive for autophagy dependent translational control of BRCA2; we believe this was due to only partial knockdown of the gene. Therefore, we have removed this data from the figure and put the graph of Atg5 null MEFs in supplementary data, which does reach statistical significance of $p < 0.05$ for BRCA2 levels. We now include new data in the supplemental figure 3 showing BRCA2 levels in wildtype MEFs following treatment with the lysosome inhibitor Bafilomycin A, which severely perturbs autophagic flux. This data supports our argument that BRCA2 levels are sensitive to autophagic capacity. We have also put the data from other HEK293T KO cell lines into the supplementary figure. Although the points do not reach statistical significance by ANOVA, we feel that because of the variability of the immunoblots, we were underpowered for our analysis. Unfortunately, collecting more samples has been impossible at this time. We have edited the text accordingly, and we thank the reviewers for the opportunity to strengthen and clarify our argument.

7. Line 213-215: “We found no significant difference between Atg12f/f and Atg12KO MEFs in either Brca2 mRNA levels (Figure 3I) or in BRCA2 protein stability or turnover following cycloheximide treatment (Figure 3J, K).” However, Fig. 3K shows that there seems a significant reduction of BRCA2 in Atg12KO ($p=0.02$) but not in the Atg12f/f control ($p=0.11$).

We apologize for not presenting our calculated p-values clearly on the graph, and we have attempted to clarify, both in the text and in the figure. The p-values on the right of the graph were calculating the significance between Atg12f/f and Atg12KO at $t=0$ and $t=6$ hr. However, we feel that it is best to remove these as they can easily be misinterpreted, and we have already

shown the statistically significant decrease in total BRCA2 levels between Atg12f/f and Atg12KO cells in Figure 3C.

8. Line 242-244: “Indeed, *Irf7*, another hit from our ribosome profiling screen showed lower protein levels in Atg12KO cells, is notable for a complex 5'UTR secondary structure 25,26 (Figure S4E).” However, Fig. S4E shows that p values are 0.52 and 0.65 between + and – Atg12, in the control and HBSS cases, respectively.

Upon further statistical analysis, one point was concluded to be an outlier by the Dixon test ($p = 0.02$). When this data point was removed, the p-value between Atg12 WT and KO in HBSS conditions is less than 0.001. As before, we are reporting all of the data points, and have specified in the figure legend how all p-values were calculated.

9. Line 261-263: “We observed increased co-location of eIF4A1 within puncta of the ACR p62/SQSTM1 in autophagy deficient cells (Figure 4G, H)”. However, Fig. 4H shows $p = 0.06$.

Upon further statistical analysis, one point was concluded to be an outlier by the Dixon test ($p = 0.03$). When this data point was removed, the p-value between Atg12f/f and KO measuring eIF4A1 colocation in p62 is $p = 0.01$. As before, we are reporting all of the data points, and have specified in the figure legend how all p-values were calculated.

10. Line 270-271: “However, p62/SQSTM1 knockdown was not sufficient to restore BRCA2 levels in Atg12KO cells (Figure 4L-N)”. However, there is lack of statistics in Fig. 4L-N.

The statistics were included in the figure legend for Figure 4N ($p = 0.76$), supporting our conclusion that knockdown of p62 was insufficient to rescue BRCA2 levels. To further elaborate, we include ANOVA statistics for figure 4M, showing that BRCA2 levels in shp62-Atg12KO are significantly lower than shNT-Atg12f/f ($p = 0.001$).

11. Line 284-286: “We therefore assayed MSI1 binding to *Brca2* by RNA immunoprecipitation and observed increased MSI1 associated with *Brca2* in the Atg12KO cells (Figure 5A).” However, Fig. 5A shows $p = 0.17$ between Atg12f/f and Atg12KO.

Upon further statistical analysis, one point was concluded to be an outlier by the Dixon test ($p = 0.007$). When this data point was removed, the p-value between Atg12f/f and KO for *Brca2* levels on MSI1 pulldown is $p = 0.006$. We are reporting all of the data points, and have specified in the figure legend how all p-values were calculated.

12. Line 287: “a modest accumulation of MSI1”. However, Fig. 5B shows $p = 0.60$ and 0.21 between + and – Atg12, in cases of the control and HBSS, respectively.

We believe that although variable, the slight increases in MSI1 that we observe is an interesting point that hints that autophagy dependent degradation of MSI1 is contributing to *Brca2* translation regulation. The lack of this data as presented would be an obvious question in the autophagy field, and therefore we feel that it is necessary to present all of the data, data points and statistics, so that the reader may be able to draw their own conclusion. We are careful however not to over-interpret these data in the results and discussion sections, and we have further specified in the text that the accumulation observed in the Atg12KO cells is not significant.

13. Line 363-365: “Immunoblotting revealed decreased BRCA2 protein levels in the kidney and cerebral cortex of Atg12KO mice compared to autophagy competent Atg12f/f controls (Figure 7E, F).” However, Fig. 7F shows that there is a significant difference between Atg12f/f and Atg12KO only in Cerebral cortex but not in Kidney.

We have observed quite a bit of variability in BRCA2 levels in control (ATG12F/F) kidney tissues, suggesting there may be other regulators of this protein in vivo. The decrease in BRCA2 in kidney is an interesting point when considered in conjunction with the other data presented in figures 7F-H, which do reach statistical significance. We are careful not to over-interpret these data in the results and discussion sections, and we have further specified in the text that the decrease observed in the Atg12KO kidney is not significant.

14. Fig. S2H (new data): It is essential to show the results at $t = 0$ to make sure the synchronization was effective equally in Atg12f/f and Atg12KO.

Unfortunately, we do not have $t=0$ data to include, and hence, we have removed Figure S2H from the manuscript. We do not feel that this significantly alters the text, as Figure S2H supported a minor point in conjunction with figures S2E, F and G.

15. Fig. 4I (new data): The essential control of eIF4A1 immunoprecipitation, the eIF4A1 staining, is missing.

We have included a total protein stain as a loading control from the pulldown showing equal levels of protein were run on the gel, which was performed at the time of the experiment. The control of an eIF4A1 blot was not performed at the time of the experiment. Due to laboratory shutdowns, it is not possible at this time to repeat the pulldown and run this eIF4A1 immunoblot control; thus, we hope that the inclusion of the total protein loading control is sufficient to satisfy the reviewer. Additionally, in a similar experiment where eIF4A1 was pulled down and RNA was isolated, in supplemental figure 4F we provide an immunoblot of eIF4A1 levels following eIF4A1 pulldown, which indicates that the pulldown is similarly efficient between Atg12f/f and Atg12KO conditions. We thank the reviewers for identifying this missing control panel.

16. I could not find the explanation of Fig. 2A-C and Fig. 5E-F in the Results section.

We have amended the text. We thank the reviewer for catching this mistake.

17. “no difference” should be “no significant difference” (lines 103, 141, and 144).

We thank the reviewers for pointing this out, and we have corrected the text.

REVIEWERS' COMMENTS:

Reviewer #3 (Remarks to the Author):

The authors addressed most of the concerns I raised in the second review. But, before publication, it is better to amend the minor points that are highlighted in red.

7. Line 213-215: "We found no significant difference between Atg12f/f and Atg12KO MEFs in either Brca2 mRNA levels (Figure 3I) or in BRCA2 protein stability or turnover following cycloheximide treatment (Figure 3J, K)," . However, Fig. 3K shows that there seems a significant reduction of BRCA2 in Atg12KO ($p=0.02$) but not in the Atg12f/f control ($p=0.11$).

We apologize for not presenting our calculated p-values clearly on the graph, and we have attempted to clarify, both in the text and in the figure. The p-values on the right of the graph were calculating the significance between Atg12f/f and Atg12KO at $t=0$ and $t=6hr$. However, we feel that it is best to remove these as they can easily be misinterpreted, and we have already shown the statistically significant decrease in total BRCA2 levels between Atg12f/f and Atg12KO cells in Figure 3C.

1. Some of the p-values are still left on the right of BRCA2 and Mcl-1 graphs (Fig. 3I). It is better to remove them ($p=0.02$, 0.80, and 0.80) as well.
2. Line 988 (J) must be (H); Line 989 (K) must be (I).

COMMSBIO19-0971C
Response to Reviewers

Response to Reviewer 3:

- 1. Some of the p-values are still left on the right of BRCA2 and Mcl-1 graphs (Fig. 3I). It is better to remove them ($p=0.02$, 0.80 , and 0.80) as well.*
- 2. Line 988 (J) must be (H); Line 989 (K) must be (I).*

As requested, we have:

- 1) removed p-values displayed on the right of the BRCA2 and Mcl-1 graphs in Figure 4I (previously Fig 3I)
- 2) we have corrected the errors in the panel callouts in the figure legend.